# Insights into HIV-1 proviral transcription from integrative structure and dynamics of the Tat:AFF4:P-TEFb:TAR complex

Ursula Schulze-Gahmen[1,2], Ignacia Echeverria[3,4,5], Goran Stjepanovic[1,2,6], Yun Bai[1,2], Huasong Lu[1,2], Dina Schneidman-Duhovny[3,4,5], Jennifer A Doudna[1,2,6,7,8], Qiang Zhou[1,2], Andrej Sali[3,4,5], James H Hurley[1,2,6]*

[1]Department of Molecular and Cell Biology, University of California, Berkeley, Berkeley, United States; [2]California Institute of Quantitative Biosciences, University of California, Berkeley, Berkeley, United States; [3]Department of Bioengineering and Therapeutic Sciences, University of California, San Francisco, San Francisco, United States; [4]Department of Pharmaceutical Chemistry, University of California, San Francisco, San Francisco, United States; [5]California Institute of Quantitative Biosciences, University of California San, Francisco, San Francisco, United States; [6]Molecular Biophysics and Integrated Bioimaging Division, Lawrence Berkeley National Laboratory, Berkeley, United States; [7]Howard Hughes Medical Institute, University of California, Berkeley, Berkeley, United States; [8]Department of Chemistry, University of California, Berkeley, Berkeley, United States

**Abstract** HIV-1 Tat hijacks the human superelongation complex (SEC) to promote proviral transcription. Here we report the 5.9 Å structure of HIV-1 TAR in complex with HIV-1 Tat and human AFF4, CDK9, and CycT1. The TAR central loop contacts the CycT1 Tat-TAR recognition motif (TRM) and the second Tat $Zn^{2+}$-binding loop. Hydrogen-deuterium exchange (HDX) shows that AFF4 helix 2 is stabilized in the TAR complex despite not touching the RNA, explaining how it enhances TAR binding to the SEC 50-fold. RNA SHAPE and SAXS data were used to help model the extended (Tat Arginine-Rich Motif) ARM, which enters the TAR major groove between the bulge and the central loop. The structure and functional assays collectively support an integrative structure and a bipartite binding model, wherein the TAR central loop engages the CycT1 TRM and compact core of Tat, while the TAR major groove interacts with the extended Tat ARM.

*For correspondence: jimhurley@berkeley.edu

## Introduction

The human immunodeficiency virus-1 (HIV-1) remains one of the world's leading health threats. In spite of advances in treatment with antiretrovirals (HAART) (*Yeni 2006*), it has not been possible to eradicate HIV-1 infection. Even under intensive antiretroviral therapy, resting T cells that harbor latent provirus integrated into their genome survive (*Archin et al., 2014*). The pool of latent virus is the primary obstacle to eradicating HIV, and thus the mechanisms by which latency is regulated are of paramount interest. The molecular mechanisms regulating HIV-1 transcription have been studied for three decades, yet renewed interest in eradicating HIV-1 infection has lent new urgency to obtaining a more complete understanding.

Proviral transcription is silenced by host epigenetic mechanisms and/or deficiency in key cofactors, and is reactivated at the level of both initiation and elongation (*Mbonye and Karn 2014*; *Zhou et al., 2012*; *Jonkers and Lis 2015*). Transcription of the HIV-1 proviral DNA is initiated by

RNA polymerase II (Pol II) binding to the HIV promoter, but stalls after a 30–50 nucleotide transcript containing the trans-activation response region (TAR) is formed. Release of the stalled Pol II is dependent on the recruitment of the host super elongation complex (SEC) (*He et al., 2010*; *Sobhian et al., 2010*) to the nascent TAR by HIV-1 trans-activator protein (Tat), a central driver of proviral transcriptional activation (*Fisher et al., 1986*).

The SEC is assembled on the AFF1 or AFF4 scaffold protein that interacts through small binding domains with the positive elongation factor b (P-TEFb), composed of CDK9 and Cyclin T1 (CycT1), the transcription factor ELL1/ELL2, and ENL/AF9 (*Chou et al., 2013*; *He et al., 2010*; *Lin et al., 2010*; *Sobhian et al., 2010*). P-TEFb facilitates promoter escape by phosphorylating two negative elongation factors (DSIF and NELF) as well as the C-terminal domain of Pol II. Overexpression of full-length AFF1, but not a fragment that only binds to P-TEFb, strongly increases Tat transactivation of transcription from the HIV promoter (*Lu et al., 2014*), indicating that productive HIV transcription requires the whole SEC.

Recent crystal structures have revealed how the intrinsically disordered P-TEFb-binding domain of AFF4 folds and binds in a hydrophobic groove on the CycT1 surface (*Schulze-Gahmen et al., 2013*; *Chou et al., 2013*). HIV-1 Tat binds to CycT1 in an extended conformation adjacent to AFF4, inter-acting with residues in the cleft between the two cyclin-fold domains of CycT1, the Tat-TAR recognition motif (TRM), and AFF4 helix α2 (*Schulze-Gahmen et al., 2014*; *Tahirov et al., 2010*; *Gu et al., 2014*). Segments from AFF4, Tat, and CycT1 TRM that are flexible in isolation come together to form a more stable structure with a positively charged surface (*Schulze-Gahmen et al., 2014*). Bio-chemical experiments have shown that both Tat and CycT1 make critical and direct contributions to TAR binding (*Calnan et al., 1991*; *Richter et al., 2002b*; *Garber et al., 1998*). In the presence of AFF4, TAR affinity to P-TEFb increases sharply (*Schulze-Gahmen et al., 2014*). The exact mechanism for the increase in affinity has been unclear, in part due to the lack of a structure of the SEC complex with HIV-1 TAR. To fill the gap in structural understanding, we determined the structure of TAR bound to Tat:AFF4:P-TEFb. We validated our low resolution crystal structure of the TAR complex with data from small angle X-ray scattering (SAXS), hydrogen deuterium exchange (HDX), selective 2'-hydroxyl acylation analyzed by primer extension (SHAPE), binding assays, and previously pub-lished biochemical data.

The structure of TAR bound to Tat:AFF4:P-TEFb reveals extensive interactions between the TAR loop and a composite protein interaction site, composed of CycT1 TRM and the Tat $Zn^{2+}$-coordina-tion loop, Tat residues 24–29. Although the Tat ARM is not visible in the electron density maps, its predicted location based on the Tat core position is close to the TAR bulge region in the crystal structure. This constraint, together with biophysical data, were used to develop an integrative model of the ARM, showing it bound through the length of the TAR major groove. AFF4 has no direct con-tacts with bound TAR in our crystal structure. Instead AFF4 is interacting with the CycT1 TRM and the second Tat $Zn^{2+}$-coordination loop. Hence, its stimulating activity on TAR binding is likely due to stabilization of the TAR interaction surface of Tat:P-TEFb by AFF4. Our results provide the first struc-tural model, albeit at low resolution, of a critical regulator of HIV-1 latency, the Tat:SEC:TAR complex.

## Results

### HDX-MS of Tat:AFF:P-TEFb with and without HIV-1 TAR

To localize the regions of the Tat:AFF4:P-TEFb complex that undergo HIV-1 TAR-induced conforma-tional changes, we performed hydrogen-deuterium exchange mass spectrometry (HDX-MS). The HDX rate of a protein region depends on its flexibility, the amount of hydrogen bonding with back-bone amides, and solvent accessibility (*Hoofnagle et al., 2003*; *Percy et al., 2012*). We compared absolute deuteron incorporation into different segments of Tat:AFF4:P-TEFb in the absence (apo complex) and in the presence of saturating amounts of HIV-1 TAR (TAR complex) (*Figure 1A*). The observed peptic peptides of the apo complex covered more than 80% of the Tat:AFF4:P-TEFb sequence (*Figure 1—figure supplement 1–3*). Our HDX data of the apo complex in solution are in good agreement with the crystal structure of Tat:AFF4:P-TEFb (*Schulze-Gahmen et al., 2014*; *Gu et al., 2014*) (*Figure 1A*). The C-terminal segment (residues 59-67) of the synthetic AFF4 peptide (32–67) shows 70% deuteration after 10 s and almost complete deuteration within 3 min

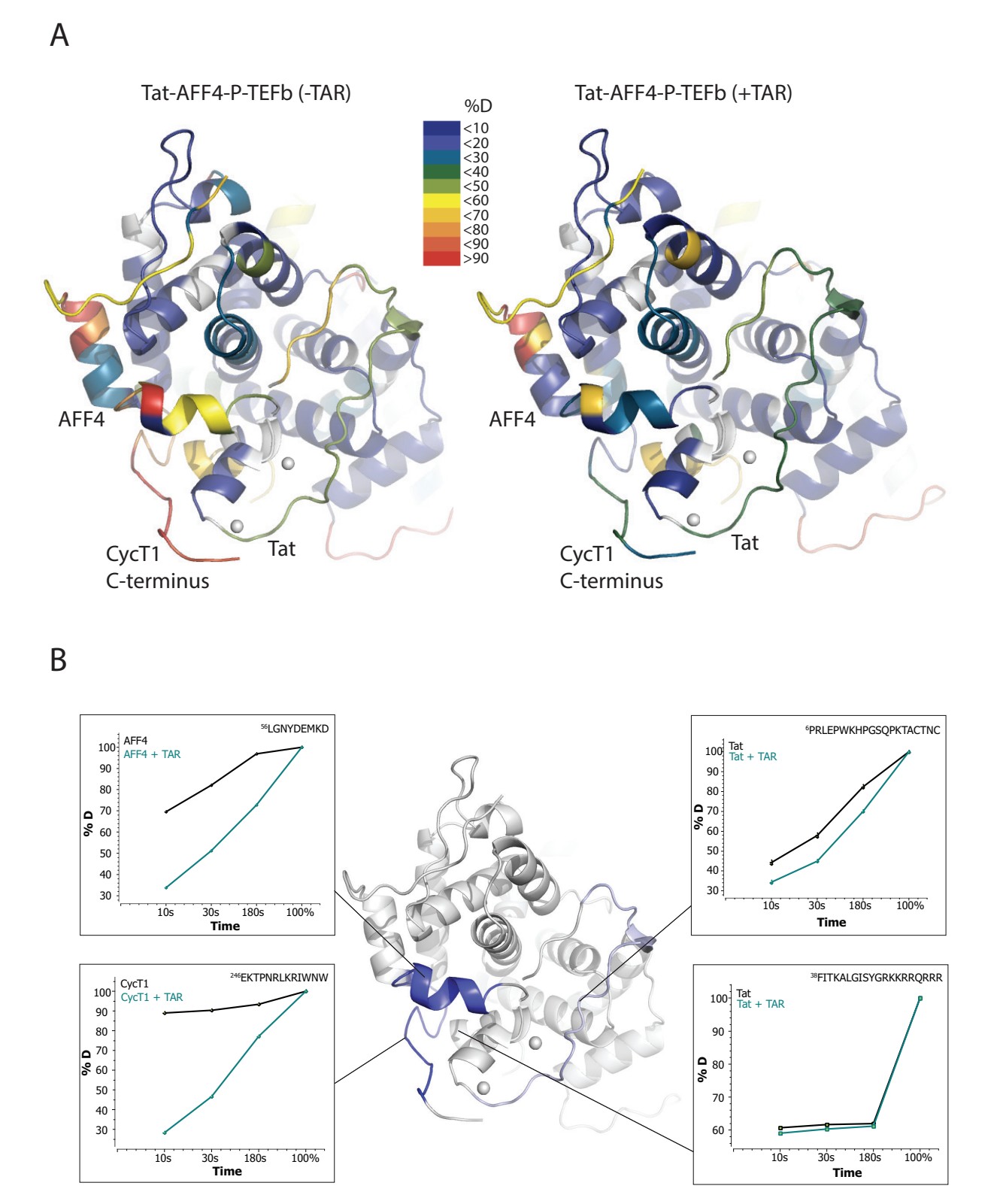

**Figure 1.** TAR-dependent changes in the Tat:AFF4:P-TEFb complex. (**A**) Structure of the Tat:AFF4:P-TEFb complex colored according to the absolute percentage of deuterons incorporated in the presence and absence of TAR after 30 s in $D_2O$. (**B**) TAR-dependent HDX kinetics of selected segments of Tat:AFF4:P-TEFb complex. Segments with more than 2.5 and 1 amide hydrogens protected in the presence of TAR at time-point 10 s are colored in dark and light blue, respectively. The HDX kinetics for these segments in the absence of TAR (black) and the presence of TAR (cyan) are shown.

*Figure 1 continued on next page*

*Figure 1 continued*

The following figure supplements are available for figure 1:

**Figure supplement 1.** Amide HD exchange results of AFF4 and TAT in the absence and presence of TAR.

**Figure supplement 2.** Amide HD exchange results of CycT1 in the absence and presence of TAR.

**Figure supplement 3.** Amide HD exchange results of CDK9 in the absence and presence of TAR.

(*Figure 1B*). Although the crystal structure indicates an α-helix in this region, high B-factors indicate high mobility and partial unfolding of this terminal helix in solution, in agreement with the high deuteration levels observed in the HDX experiment. The Tat-TAR recognition motif (TRM; residues 246 to 262 in CycT1, (*Das et al., 2004*; *Garber et al., 1998*) also showed almost complete deuteration within 10 s in the apo complex (*Figure 1B*), which is consistent with multiple conformations for the TRM in the crystal structure (*Schulze-Gahmen et al., 2014*).

Compared with the apo complex, we observed strong protection for the CycT1 TRM and AFF4 helix α2 (residues 59–65) after TAR binding (*Figure 1B*, dark blue segments) (*Figure 1—figure supplement 1–3*). These two segments are adjacent to each other in the crystal structure of Tat:AFF4:P-TEFb, consistent with their mutual cooperation in forming the TAR binding site. In contrast to CycT1 and AFF4, Tat exhibited a lesser degree of protection, localized to the N-terminal segment (residues 1-33) (*Figure 1B*, light blue segments). Surprisingly, the Tat ARM did not show any significant protection in HDX after TAR binding despite its well-established role in TAR binding (*Weeks and Crothers, 1991*; *Calnan et al., 1991*) (*Figure 1B*). This observation indicates that Tat ARM–TAR interactions are mediated primarily by amino acid side chains, without extensive hydrogen bonding and solvent exclusion of main-chain amide protons.

## SHAPE mapping of the Tat:AFF4:P-TEFb complex on the HIV 5'UTR

We performed SHAPE analysis (*Merino et al., 2005*) to map the interactions of the Tat:AFF4:P-TEFb complex on the HIV 5'UTR, which includes the TAR sequence at its 5' end. Experiments were done using the 344 nt-long 5' end of the HIV genome (*Heng et al., 2012*) in the presence and absence of the Tat-AFF4-P-TEFb complex. Upon protein complex binding, the major SHAPE changes cluster in the 5' TAR region of the 5'UTR (*Figure 2A,B*) with significant SHAPE changes observed around the bulge region at A17, the bulge region A22-U25, as well as the loop region C30-A35. As expected, no significant reactivity changes were observed in the regions 3' to TAR. The SHAPE reactivity for some nucleotides in bulge A22-U25 and loop C30-A35 is extremely high (above 3) in the absence of Tat:AFF4:P-TEFb and remains high even after Tat:AFF4:P-TEFb binding. TAR binding reduces SHAPE reactivity by 0.3-0.4 and 0.2-1.2 for nucleotides in the bulge and loop region, respectively. The pattern of changes in SHAPE reactivity for TAR after binding the SEC is significantly different from changes observed after TAR binding to a Tat peptide covering the ARM region (*Kenyon et al., 2015*). This difference is expected because of additional contacts between CycT1 and TAR, which lead to higher affinity and more specific interactions between TAR and the SEC (*Richter et al., 2002b*; *Zhang et al., 2000*) compared to interactions between TAR and a small arginine-rich peptide from Tat. Because reduction in SHAPE activity correlates with reduced flexibility of the respective RNA regions (*McGinnis et al., 2012*), the results for the TAR-SEC complex indicate a stabilization of the bulge and loop structure through direct contacts between TAR and the SEC (*Figure 2*).

## SAXS of the SEC complex with TAR

SAXS data were collected for Tat:AFF4:P-TEFb complex in the presence and absence of TAR (*Figure 3A*). A comparison of SAXS data reveals significant differences in the $P(r)$ function (*Figure 3B*), radius of gyration ($R_G$), and the Porod exponent (*Table 1*). The $R_G$ of the TAR complex increases by 2.8 Å upon TAR binding. The Porod exponent, in contrast, is smaller for the TAR complex than the apo complex. The Porod exponent is a measure of the compactness of a molecule,

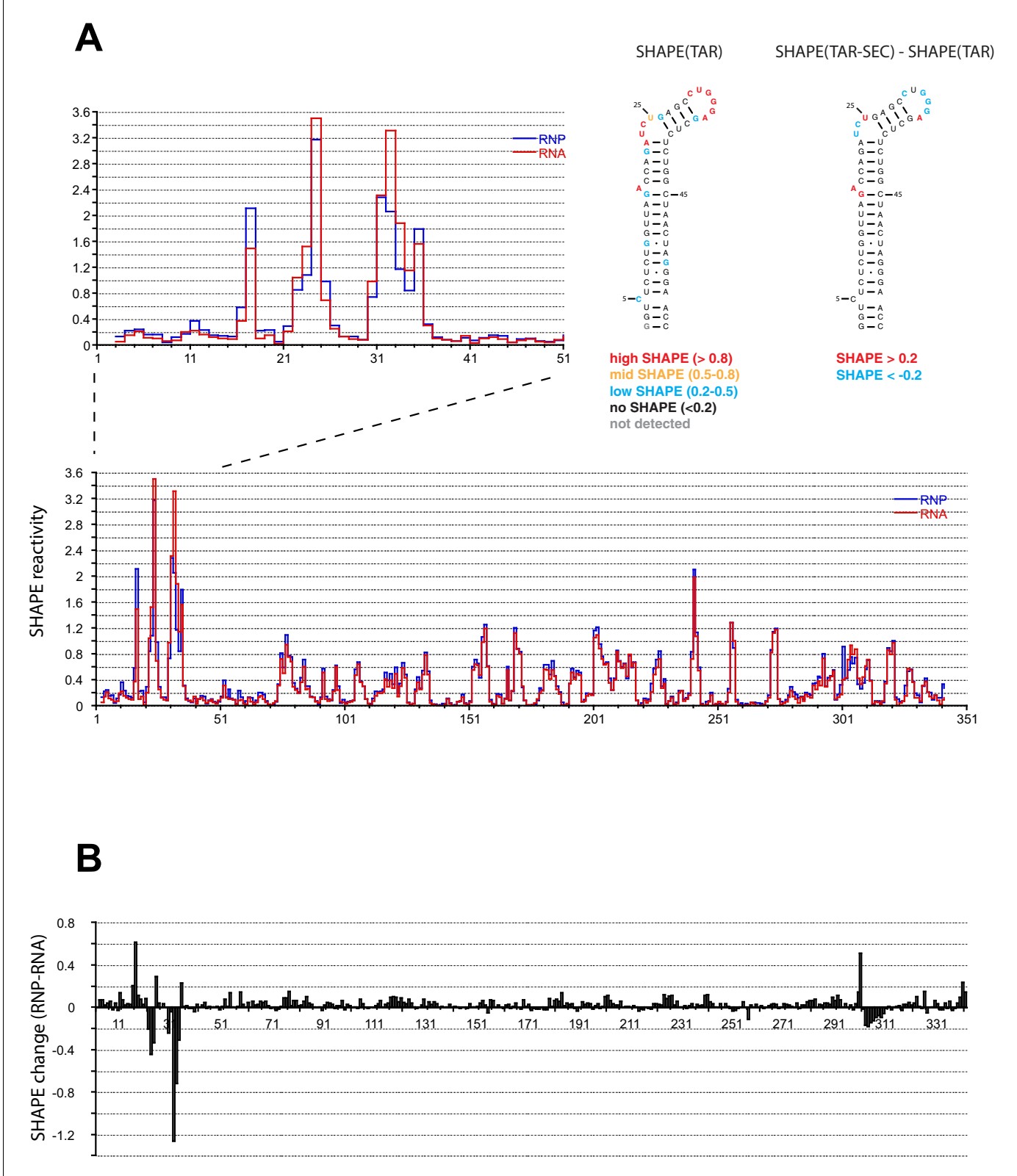

**Figure 2.** SHAPE profiles of the HIV-1 5′UTR in the presence and absence of the SEC. (**A**) SHAPE reactivity of the entire HIV-1 5′UTR either alone or in complex with the SEC are shown in red and blue, respectively. SHAPE reactivity profiles are normalized using the 2/8 rule, where the top 2% of reactive nucleotides are excluded and the average of the next 8% of reactive nucleotides is used as the normalization factor. The left insert is a zoomed-in view

*Figure 2 continued on next page*

*Figure 2 continued*

of the TAR region. The right insert shows the secondary structure of TAR color-coded by the SHAPE profile of the TAR region alone and the SHAPE changes in the TAR region upon SEC binding. (**B**) SHAPE changes of the HIV-1 5′UTR upon SEC binding.

The following figure supplement is available for figure 2:

**Figure supplement 1.** Control trace for reverse transcription reaction with HIV-1 5′UTR without SHAPE reagent.

with a larger exponent indicating a more compact structure. The TAR complex appears slightly less compact than the apo structure, which agrees with the differences in the shape of the *P*(r) function.

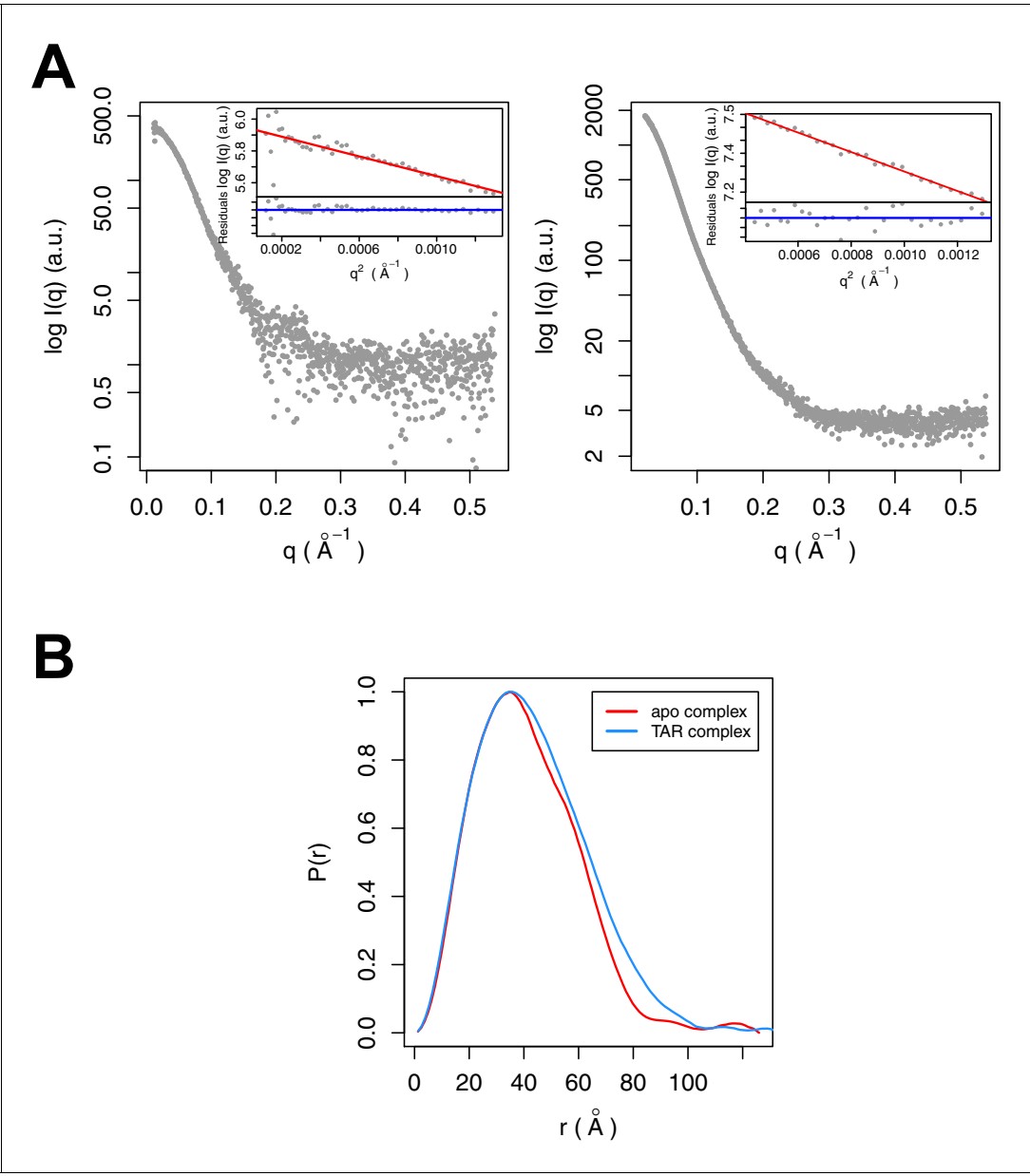

**Figure 3.** SAXS analysis of the apo and TAR-bound Tat:AFF4:P-TEFb complexes. (**A**) Experimental intensities for the apo complex (left) and the TAR complex (right). Inserts in the top right corner show the Guinier plots for each data set. (**B**) The distance distribution plots were computed from the experimental data and normalized to the maximum value of unity.

**Table 1.** Overall parameters of Tat:AFF4:P-TEFb complexes determined by SAXS

| Complex | $R_G$, Å[*] | $D_{max}$, Å[*] | Porod exponent[*] |
|---|---|---|---|
| Tat:AFF4:P-TEFb | 31.2 | 126 | 3.6 |
| TAR:Tat:AFF4:P-TEFb | 33.9 | 134 | 3.4 |

[*] Parameters were determined using the program SCÅTTER (*Förster et al., 2010*).

The *P*(r) curve for the TAR complex is shifted to larger radii, with especially large differences in the distance distributions at r values between 70 and 130 Å.

## Crystal structure of the SEC complex with TAR

To clearly define the interactions between TAR and Tat:P-TEFb, and to determine if direct interactions existed between TAR and AFF4, we solved the crystal structure of the TAR:Tat:SEC complex at 5.9 Å resolution (*Table 2*). The structure was solved by molecular replacement with one copy of the Tat:AFF4:P-TEFb complex (PDB ID 4OGR) in the asymmetric unit. After rigid-body and torsion angle molecular dynamics refinement with deformable elastic network (DEN) restraints (*Table 2*) (*Schröder et al., 2010*; *Brunger et al., 2012*) strong density for the TAR RNA became visible in the Fo-Fc electron density map (*Figure 4—figure supplement 1*) which allowed placement of one TAR molecule (PDB ID 1ARJ) into the asymmetric unit. The complex was further refined to R/R$_{free}$ = 0.211/0.311 with good geometry (*Table 2*) and an excellent fit to the density (*Figure 4—figure supplement 2*).

The extended TAR loop region 29-33 contributes the main interactions with the SEC. It interacts with the protein complex surface composed of CycT1 TRM and the Tat cysteine-rich region, especially the Zn$^{2+}$-coordinating loop (Tat 24-29). This structure rules out direct interactions between TAR and AFF4 α2. TAR loop nucleotides G32 and G33 abut the Tat Zn$^{2+}$-coordination loop (24-29) and CycT TRM, residues 259-261. The backbone phosphates of nucleotides 30–32 contact basic residues in the CycT TRM region (*Figure 4*). This orientation places the N-terminus of the Tat ARM (Arg49) at the widest point in the TAR major groove, which is 13 Å wide between nucleobases C24 and G34. However, there is not interpretable electron density present for residues in the Tat ARM region at the limited resolution of this crystal structure. This seems consistent with the lack of protection of this region seen in the HDX experiment (*Figure 1B*), but it is important not to overinterpret this observation given the limitations of the 5.9 Å map.

The structure of the TAR complex with Tat:AFF4:P-TEFb shows no direct contacts between AFF4 and TAR. Instead, AFF4 helix α2 and its N-terminal loop connection to helix α1 are interacting with CycT1 TRM and Tat helix α1. The interaction network between these three protein regions most likely stabilizes the protein interaction surface for TAR and indirectly increases Tat:SEC affinity for TAR in the presence of AFF4 (*Schulze-Gahmen et al., 2014*).

## Modeling of Tat ARM and CycT1 TRM

The Tat ARM and CycT TRM are key components for TAR binding by the SEC. Since the crystal structure shows no electron density for Tat ARM and the molecular replacement model for CycT1-TRM segment adopts multiple conformations we modeled the Tat ARM and refined the structure of the CycT1-TRM segment using the MODELLER 9.15 loop modeling protocol (*Šali and Blundell, 1993*; *Fiser and Šali, 2003*). Using hierarchical clustering and DOPE scoring (*Shen and Sali, 2006*) we obtained five best scoring clusters with precisions between 2.5 and 6.3 Å (*Figure 5*). The computed small angle scattering fits well with SAXS data obtained from the TAR complex in solution with χ=0.86 (*Figure 5—figure supplement 1*). The CycT1 TRM segment is anchored by CycT1 Cys261, which coordinates one of the Tat Zn$^{2+}$ ions and displays structural variability in the center part of the TRM (*Figure 5B*). The Tat ARM region is N-terminally anchored by Tat Tyr47, which binds to a pocket on the CycT1 surface and interacts with CycT1 residues Leu44, Asp47, and Cys111 (*Figure 5C*). From there, the ARM (residues 49-57) extends into the major groove of TAR stretching from the TAR bulge towards the phosphate backbone opposite the bulge. Basic side chains in the Tat ARM are in positions where they can form previously described interactions with the TAR G26

**Table 2.** X-ray data collection and refinement statistics for Tat:AFF4:P-TEFb with TAR21.

| Data collection | |
|---|---|
| Space group | P3$_2$21 |
| Cell dimensions: $a$, $b$, $c$ | 146.87, 146.87, 103.75 |
| Resolution (Å)[*] | 50.0-5.9 (6.1–5.9) |
| Unique reflections[*] | 3607 (339) |
| I/σ(I)[*] | 10.22 (0.35) |
| R$_{sym}$ (%)[*] | 17.8 (>100) |
| CC$_{1/2}$ high resolution shell | 0.24 |
| Completeness (%)[*] | 100.0 (100.0) |
| Redundancy[*] | 9.68 (10.25) |
| Twin fraction[†] | 0.03 |
| Temperature (K) | 100 |
| Mosaicity (°) | 0.175 |
| | |
| **Refinement** | |
| Resolution (Å) | 50.0-5.9 |
| No. reflections | 3595 |
| DEN refinement parameter[§] | γ=0, w$_{DEN}$=300 |
| R$_{work}$/R$_{free}$ | 0.223/0.315 |
| Average B-factor (Å$^2$) | 400 |
| R.m.s. deviations | |
| Bond lengths (Å) | 0.0027 |
| Bond angles (°) | 0.81 |
| Ramachandran plot[#] | |
| Favored (%) | 92.01% |
| Allowed (%) | 6.43% |
| Disallowed (%) | 1.57% |

[*]Values in parentheses are for the highest resolution shell.
[†]Value from Ctruncate (see **Table 2—source data 1**)
[§]parameter for DEN refinement in CNS (see **Table 2—source data 1**)
[#]Values from COOT.

**Source data 1.** Diffraction data and refinement analysis. Twinning analysis by L-test in Ctruncate (**A**) and Xtriage indicate no twinning. Blue curve is acentric untwinned data, green curve acentric twinned data, and red curve observed data from Tat:AFF4:P-TEFb-TAR co-crystal. (**B, C**) Optimization of DEN refinement parameters using the web service for low resolution crystal structure refinement (https://portal.sbgrid.org/d/apps/den/). (**D**) Log-file of twinning analysis in CNS.

**Source data 2.** RSCC and RSR=sum|r-r_calc|/sum[r+r_calc] using 2fofc map and D*Fcalc.

nucleobase, and phosphates P22, P23, and P40 in the TAR backbone (*Puglisi et al., 1993*; *Aboul-ela et al., 1995*).

## Effect of AFF4 mutants on TAR binding and transcriptional activation

The strong protection of AFF4 helix α2 (residues 58-66) observed in HDX experiments could be due to direct interactions between the AFF4 helix 2 and TAR or to an indirect stabilization of the helical structure of AFF4 after TAR binding. To distinguish between these two possibilities, we explored the

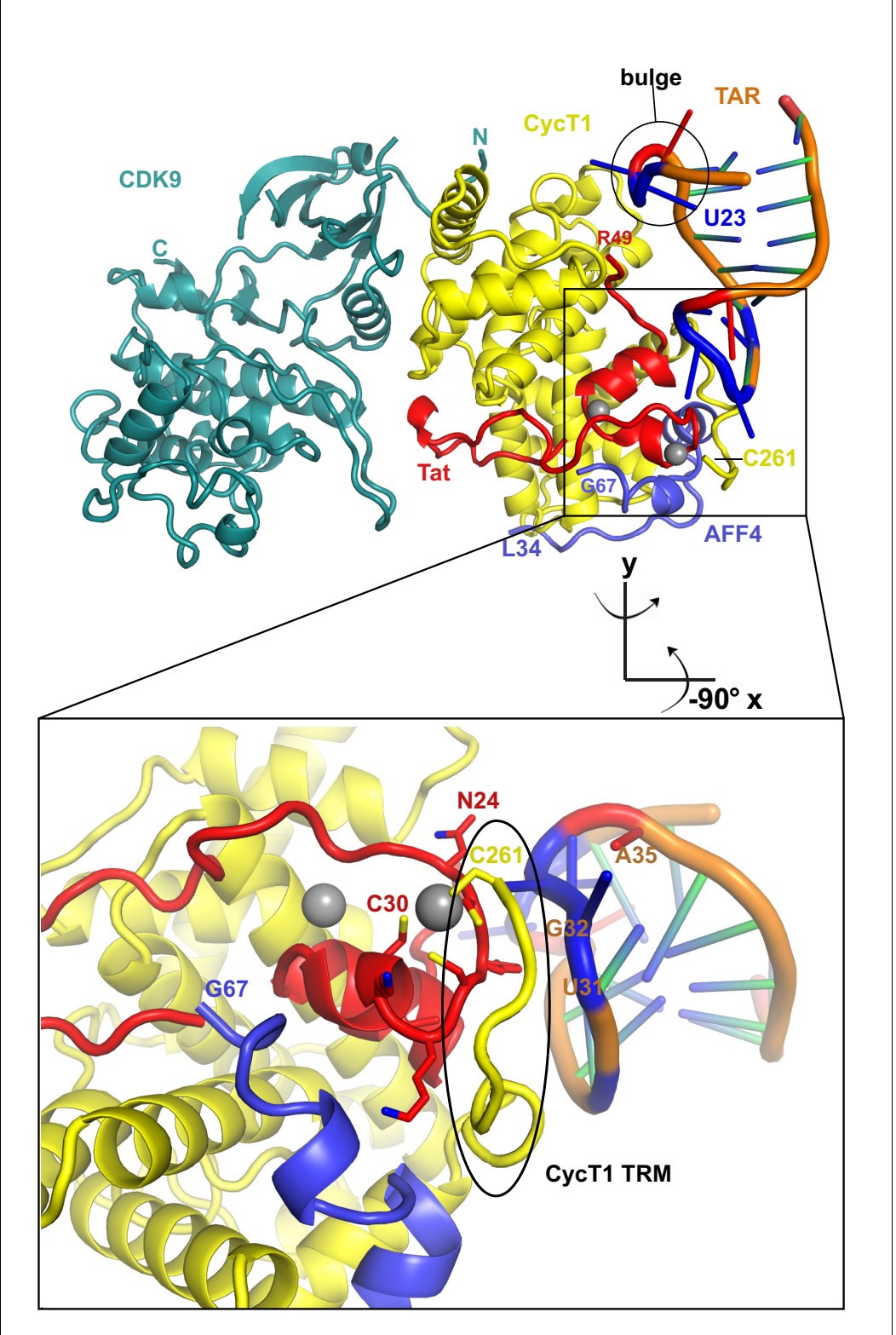

**Figure 4.** Ribbon diagram of the crystal structure of HIV-1 TAR in complex with Tat:AFF4:P-TEFb. The top panel shows an overview of the complex with the TAR loop next to CycT1 TRM (yellow) and Tat (red), and the TAR bulge close to Tat R49, the start of the Tat ARM. $Zn^{2+}$ ions are shown as grey spheres. The close-up view below highlights the interactions of the $Zn^{2+}$ binding loop and CycT1 TRM with the TAR-loop and the role of AFF4 (blue) in indirectly stabilizing the Tat-CycT1 surface involved in TAR interactions. TAR nucleotides with reduced SHAPE

*Figure 4 continued on next page*

*Figure 4 continued*

activity (< −0.2) in the presence of SEC are colored blue, those with increased SHAPE activity (> 0.2) are colored red.

The following figure supplements are available for figure 4:

**Figure supplement 1.** Fo-Fc density map (2σ) after molecular replacement and refinement with Tat:AFF4:P-TEFb only.

**Figure supplement 2.** 2Fo-Fc electron density map for the refined Tat:AFF4:P-TEFb complex with TAR.

effect of mutations in the surface-exposed side of AFF4 helix α2 on TAR binding. We performed electrophoretic mobility shift assays (EMSA) with labeled TAR and Tat:P-TEFb complex in the presence of excess WT or mutant AFF4 fragment (residues 2-73) (*Figure 6A*). The single site mutants

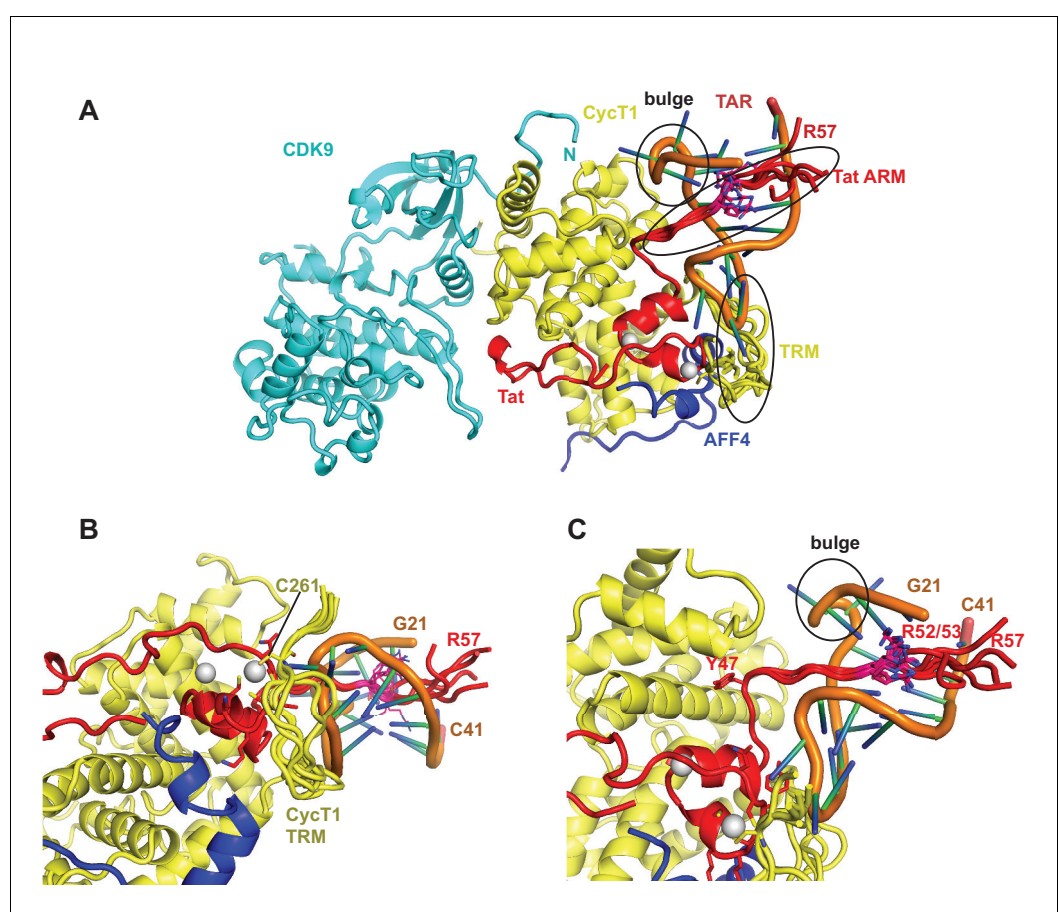

**Figure 5.** Ribbon diagram of the crystal structure of the TAR-SEC complex including modeled Tat ARM and CycT1 TRM regions. (**A**) Overview of the whole complex with multiple conformations for ARM and TRM, using the same color coding as in *Figure 4*. (**B**) Closeup of the interactions between TAR–loop and CycT1 TRM. The complex is rotated 90 degree around x relative to (**A**). (**C**) Tat ARM is N-terminally anchored by Tat Y47 to CycT1. The extended ARM is located in the TAR major groove, positioning R52 and R53 close to the TAR bulge and TAR phosphate backbone.

The following figure supplement is available for figure 5:

**Figure supplement 1.** Fit of integrative model and SAXS profile.

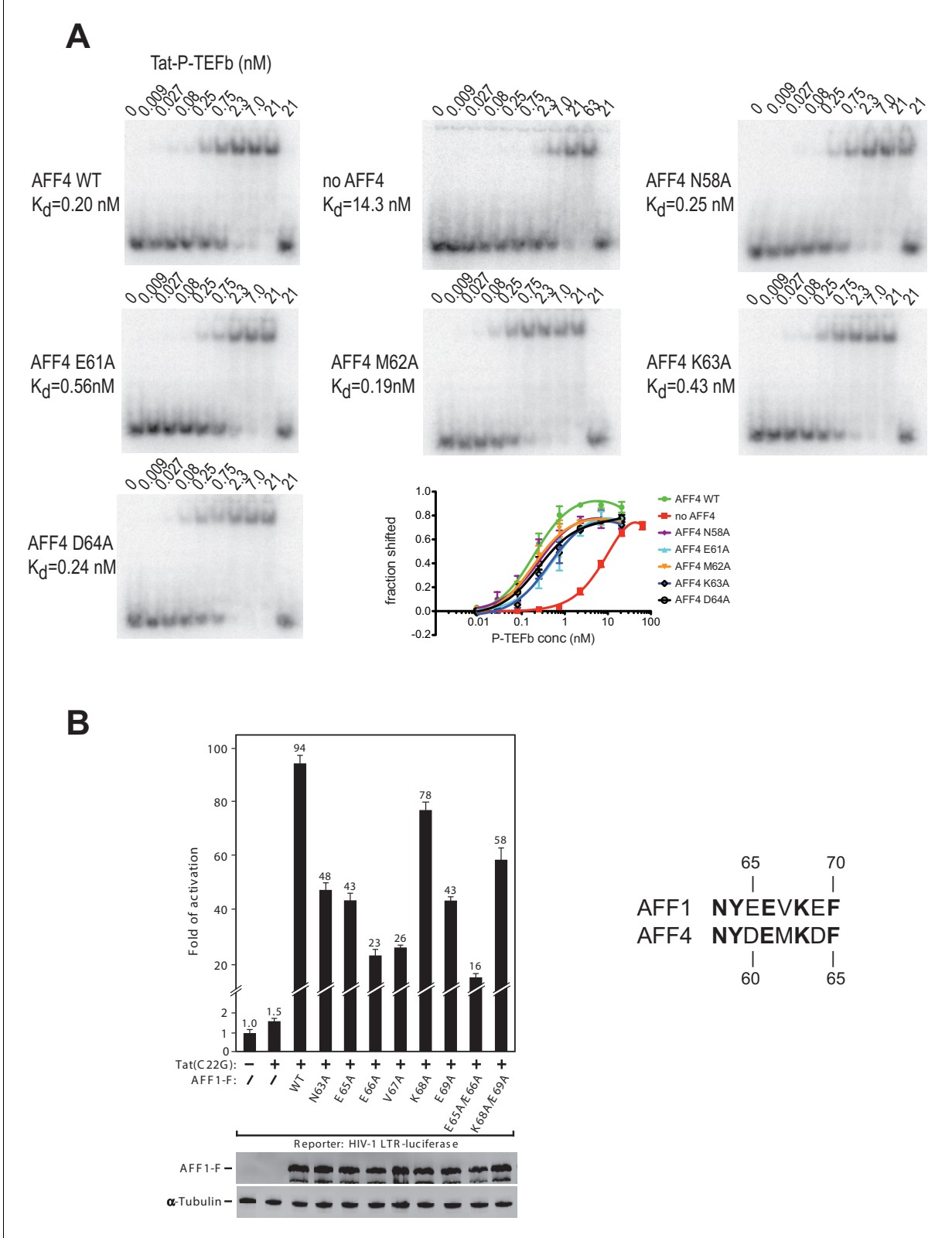

**Figure 6.** AFF4 mutants in helix α2 affect TAR binding. (**A**) Electrophoretic mobility shift assays with $^{32}$P-labeled TAR and increasing concentrations of Tat:P-TEFb in the absence or presence of excess AFF4 WT fragment (resides 2–73) or mutant AFF4 fragment. In the absence of AFF4, affinity for TAR is reduced 50-fold. Single site mutants E61A and K63A reduce TAR affinity 3 and 2-fold, respectively. (**B**) HeLa-based NH1 cells containing the integrated HIV-1 LTR-luciferase reporter gene were transfected with the Tat(C22G)- and/or AFF1-expressing construct as labeled. Luciferase activities were

*Figure 6 continued on next page*

*Figure 6 continued*
measured in cell extracts, with the level of activity detected in cells transfected with an empty vector (-) set to 1. The error bars represent ± one standard deviation from three independent measurements.

AFF4 E61A and AFF4 K63A show a three-fold and two-fold reduction in TAR binding, respectively, while the effect of the other mutations is negligible. The gel shift in the absence of AFF4 shows a 50-fold increased $K_d$ for TAR binding by Tat:P-TEFb ($K_d$ = 19 nM) compared to the gelshift in the presence of WT AFF4 ($K_d$ = 0.26 nM). The small effect of surface mutations in AFF4 helix $\alpha2$ on TAR binding indicates that instead of direct interactions with TAR structural stabilization of the combined interface formed collectively by Cyc T1:Tat:AFF4 (*Schulze-Gahmen et al., 2014*) is the major role of AFF4 $\alpha2$.

To examine the AFF1/4 contribution to Tat-dependent HIV-1 transcription, we probed the effect of alanine substitutions on Tat-dependent HIV-1 transcription in HIV-1 LTR-driven luciferase expression. These assays were performed with the AFF1 scaffold protein, because Tat has a stronger effect on HIV transcription with AFF1 than with AFF4 (*Lu et al., 2015*). In addition, the assay used Tat (C22G), a Tat mutant that lacks an important Cys-$Zn^{2+}$ ion coordination site required for wild-type Tat activity (*Garber et al., 1998*). It is thus highly sensitive to the AFF-mediated increase in affinity for P-TEFb (*Lu et al., 2014*). While wild-type AFF1 strongly supported the activation of HIV-1 LTR by Tat(C22G), all mutants displayed decreased activity, with 4-6-fold decreases observed when Glu65 (AFF4 Asp60), Glu66 (AFF4 Glu61), and Val67 (AFF4 Met62) were either singly or doubly mutated (*Figure 6B*). The combined results from EMSAs and transcription assays are consistent with the structural inference that AFF1/4 helix $\alpha2$ plays a significant role in mediating TAR binding and Tat-transactivation, although the crystal structure shows no direct contact between them.

## Effect of TAR mutants on SEC binding

Previous studies have identified several TAR mutations that severely reduce transactivation activity of TAR and TAR binding to a Tat-peptide spanning the ARM region of Tat (*Churcher et al., 1993*). Three mutants with strongly reduced transactivation activity include $G_{21}$-$C_{41}\rightarrow$ AU, $A_{22}$-$U_{40}\rightarrow$ UA, and $U_{23}\rightarrow$C, which are located just below and in the TAR bulge. To further evaluate AFF4 function in TAR binding we determined binding affinities for WT-TAR and TAR-mutants to Tat:P-TEFb with or without AFF4 (*Figure 7*) in electrophoretic mobility shift assays (EMSAs). In the absence of AFF4, all three TAR-mutants showed strongly reduced binding affinity compared to the WT TAR. However, in the presence of AFF4, the affinity of WT TAR was increased 50-fold to 0.2 nM, and binding of the TAR-mutants was rescued to a 2–10 fold increased $K_D$ compared to WT TAR (*Figure 7*). The potent rescue of bulge mutants by addition of AFF4 is consistent with the placement of the bulge distal to the Tat zinc-binding loops. The picture that emerges is of a bipartite mode of TAR binding to the SEC. Nucleobases C29-G34 interact directly with the well-ordered Tat-SEC epitope consisting of the Tat zinc-binding loops and the adjacent N-terminal portion of the CycT1 TRM. This well-ordered region is the part that is scaffolded by AFF4. The remaining nucleobases of the Tat-SEC binding site on TAR, including the bulge, interact with extended regions of the C-terminal part of the CycT1 TRM and with the Tat ARM.

## Discussion

Most of what we know about how HIV-1 TAR binds to HIV-1 Tat comes from some dozens of NMR structural studies of TAR in complex with argininamide or various peptide models of Tat (*Aboul-ela et al., 1995*; *Puglisi et al., 1992*; *Puglisi et al., 1993*; *Long and Crothers 1999*; *Davidson et al., 2011*, *2009*). Tat does not function alone in HIV-infected human cells. Rather, it hijacks the host SEC, consisting of CDK9 and Cyclin T1 (P-TEFb), AFF1/AFF4, ELL1/ELL2, and ENL/AF9 (*Sobhian et al., 2010*; *He et al., 2010*; *Luo et al., 2012*; *Lu et al., 2013*). It has been known for nearly 20 years that the interaction of Tat with P-TEFb alters the specificity of its interaction with TAR. Thus a key goal in the field has been to understand the structural basis for TAR binding to Tat in the context of the SEC. Yet it has proved challenging to obtain structural data on HIV TAR in the context of the functional human Tat-SEC complex.

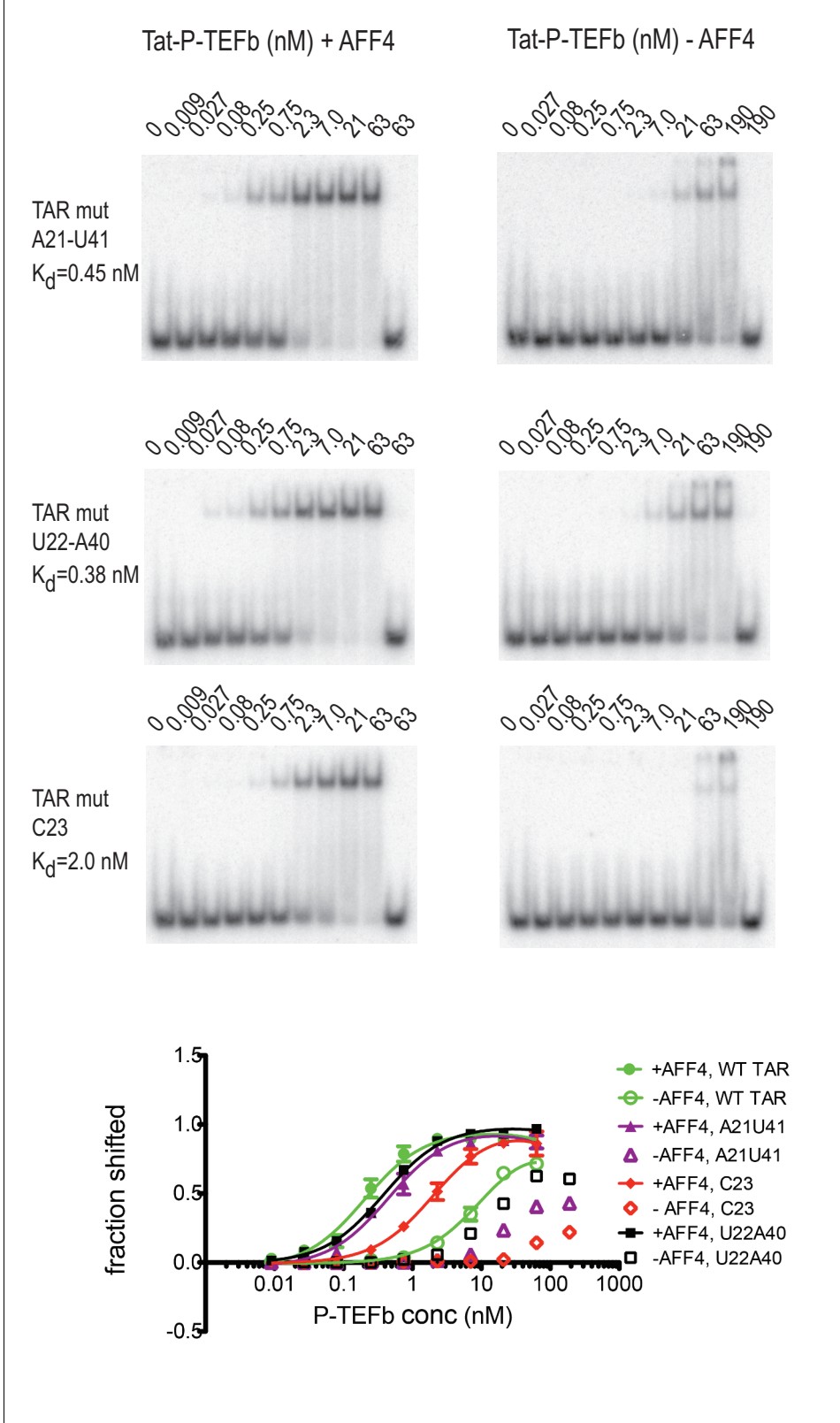

**Figure 7.** Effect of TAR mutations on Tat:P-TEFb binding in the absence and presence of AFF4. Electrophoretic mobility shift assays with $^{32}$P-labeled TAR were performed as in *Figure 6*. TAR mutants G21–C41→A21–U41, A22–U40 → U22–A40, and U23C (*Churcher et al., 1993*) show large defects in binding to Tat:P-TEFb, but can be rescued in the presence of AFF4. Experiments were performed in triplicate. The error bars represent the standard deviation from three independent measurements.

We have now determined the first crystal structure of a TAR:Tat:SEC complex to 5.9 Å resolution (*Figure 4*). Despite the resolution, the maps are adequate to rule out alternative orientations of TAR. We applied a battery of biophysical studies to support and validate the low-resolution crystal structure. These data elucidate important dynamical changes, especially in AFF4, that are not evident from the crystal structure alone. The Tat ARM region was missing in the crystal structure. The Tat ARM, as well as the partially disordered CycT1 TRM segment, were modeled as loop ensembles inserted into the crystal structure (*Figure 5*). In agreement with previous biochemical data (*Wei et al., 1998*; *Garber et al., 1998*; *Richter et al., 2002a*, *2002b*), HIV-1 TAR binds with the loop region interacting with CycT1-TRM and the $Zn^{2+}$-coordinating core region of Tat, while the TAR bulge is positioned close to the expected location of Tat ARM (*Figure 4*, *5*). Results from HDX experiments for the ARM (*Figure 1*) indicate a high flexibility for the Tat ARM even after binding to TAR. Indeed, the Tat peptide 38-57 exchanged 60% of its protons within 10 s, whether TAR was bound or not. This region of Tat is visualized in the crystal structure through residue 49, and the amides are protected by helical hydrogen bonds through residue 45. Thus 37% of the peptide is protected from exchange, closely matching the 40% that is involved in helical amid hydrogen bonding in the crystal structure. This supports that the ARM main-chain is extended and exposed when bound to TAR.

The five best-scoring Tat ARM structures in integrative modeling are all in an extended conformation in the TAR major groove (*Figure 5C*), which allows the basic residues in the ARM to make contacts with bases in the TAR bulge and with the phosphate backbone. In contrast, the TAR central loop binds to a more structured protein surface, composed of the CycT1 TRM and the second Tat $Zn^{2+}$ ion coordinating loop, Tat residues 24-29. Although AFF4 has no direct interactions with TAR, residues in helix α2 bolster Tat helix α2 and the $Zn^{2+}$ coordinating loop, and the CycT1 TRM. This network of AFF4 interactions, which stabilizes CycT1 and Tat conformations in the TAR interacting region, is the likely cause for the 50-fold increased binding affinity of Tat:P-TEFb for TAR in the presence of AFF4 (*Figure 6*, *7*).

Consistent with the bipartite mode of binding suggested by the crystal structure and dynamics data, AFF4 is able to rescue binding of TAR mutants in the bulge and the stem region just below the bulge (*Figure 7*). This rescue effect is attributed to the supporting and stabilizing role of AFF4 on the TAR interacting surface composed of CycT1 TRM and Tat loop, 24-29. Our structural and binding studies show that TAR loop interactions with the CycT1-Tat core interface play a critical role for Tar binding and in the presence of AFF4 can even compensate for reduced binding between Tat ARM and TAR bulge mutants. The crystal structure is in good agreement with the results of our SHAPE experiments with HIV-1 5'UTR with or without Tat:AFF4:P-TEFb, as SHAPE-protected nucleotides C30, G32, G33, and G34, shown in blue in *Figure 4B*, are all in close contact with CycT1 TRM or Tat in the crystal structure.

Our crystal structure of HIV-1 Tat:AFF4:P-TEFb with TAR differs significantly from the crystal structure of an equine CycT1-EIAV Tat fusion co-crystallized with EIAV TAR (*Anand et al., 2008*). In that structure, the Tat activation domain, which directs the incorporation of Tat into the SEC is disordered, and AFF1/4 is not present. The Tat ARM is found in a helical structure bound to the cyclin box and interacting with the EIAV TAR in the major groove and loop region. This complex structure is difficult to reconcile with the HDX data for the Tat ARM and CycT TRM, as well as with the critical role of human CycT1 residues in the TRM for TAR binding (*Garber et al., 1998*). This may reflect differences between EIAV and HIV-1 and/or the disorder of the Tat $Zn^{2+}$ domain in this structure.

While it is exciting to have a low-resolution structural view of this elusive complex, it is important to bear in mind its limitations. Side-chain conformations and details of extended main-chain segments are not defined in a 5.9 Å resolution crystal structure, nor by global information from SAXS or by the main-chain directed HDX experiment. SHAPE probes the RNA backbone reactivity, but it does not provide detailed information about the conformations of the nucleobases. This low-resolution structure provides a useful starting model until a high-resolution structure can be determined.

Considerable effort has been expended over the past two decades in the design of Tat antagonists, intended to inhibit HIV-1 transcriptional elongation. This continues to be an important goal, but more attention is now focused on promoting proviral transcription to deliberately reactivate latent virus and so eradicate latent viral reservoirs. Thus, it is at least as interesting to develop Tat-TAR interaction agonists. Whether the therapeutic goal is activation or inhibition, it is important to

bear in mind that Tat:TAR functions in the context of the SEC. The insight into the assembly of this complex will be helpful in pursuing either one.

## Materials and methods

### Protein expression and synthesis

P-TEFb and P-TEFb-Tat$_{1-57}$ were expressed in High 5 insect cells using recombinant baculovirus infections. We co-expressed human CDK9 1–330 and human cyclin T1 1–264 with and without HIV-1 Tat 1–57. Baculovirus generation and High 5 cell infections were described in detail previously (*Schulze-Gahmen et al., 2013*). AFF4 fragments 2–73 with a N-terminal TEV-protease-cleavable His-tag were expressed in *E. coli* (*Schulze-Gahmen et al., 2013*). The AFF4 peptide 32-67 with acetylated and amidated termini was synthesized at the University of Utah DNA/Peptide Facility.

### TAR RNA

A synthetic TAR fragment encompassing nucleotides 18–44 (TAR27) or 21–41 (TAR21) were purchased from IDT (San Diego, CA, USA). The RNA was annealed at 0.1 mg/ml in 20 mM Na HEPES pH 7.3, 100 mM KCl, 3 mM MgCl$_2$. Best results were obtained by heating the RNA at 75°C for 2 min, followed by rapid cooling on ice. The purity of the RNA, analyzed by denaturing and native 10% polyacrylamide gel electrophoresis, was at least 95%.

### Protein purification

Tat:P-TEFb and AFF4$_{2-73}$ were purified separately following procedures described recently (*Schulze-Gahmen et al., 2013*). Tat-P-TEFb and AFF4$_{2-73}$ (or AFF4$_{32-67}$) were combined at a 1:1.4 (mol/mol) ratio, concentrated to 0.6 ml, and injected onto an analytical Superdex S200 size exclusion column equilibrated with 25 mM Na-HEPES pH 7.4, 0.2 M NaCl and 1 mM DTT. To purify the Tat:AFF4:P-TEFb complex with TAR, synthetic TAR was added in small molar excess to the protein complex prior to purification over an analytical Superdex S200 column. The center fractions of the eluted S200 peak were used for SAXS data collection. Fractions with base line absorption, collected later in the elution process, were used to measure background diffraction for the SAXS experiment.

### Crystallization and structure determination

Crystals of the TAR:Tat:SEC complex grew easily under low salt conditions but diffracted very poorly. Optimization of the TAR construct used for crystallization eventually resulted in needle shaped crystals diffracting to 5.9 Å resolution. Purified Tat$_{1-57}$:AFF4$_{32-67}$:P-TEFb was combined with the annealed TAR21 fragment, nucleotides 21-41, at a 1: 1.3 (mol/mol) ratio and concentrated to 7 mg/ml in 25 mM HEPES pH 7.3, 0.2 M NaCl, 0.05M KCl, 0.1 M Ammonium sulfate, 3 mM MgCl2, 0.5 mM TCEP. Crystals were grown in sitting drops from 0.8 ul protein-TAR complex combined with 0.5 ul reservoir solution. The drops were equilibrated against 50 mM Tris 8.5, 0.2M Ammonium Acetate, 6 mM MgCl$_2$, 8% PEG 4K at 18°C. Single needle-shaped crystals grew to a size of about 0.05 mm x 0.05 mm x 0.25 mm.

Crystals were soaked in 0.1 M HEPES pH 8.0, 50 mM NaCl, 50 mM Ammonium Acetate, 6 mM MgCl$_2$, 15% PEG 4K, 30% glycerol for cryoprotection and flash frozen in liquid nitrogen. X-ray data were collected at Beamline 8.3.1 at the Advanced Light Source at the Lawrence Berkeley National Laboratory (*MacDowell et al., 2004*) using a Pilatus 3 6M detector (Dectris AG, Baden-Dättwil, Switzerland). The reflections were processed using XDS/XSCALE (*Kabsch 2010*) (*Table 2*). The R$_{sym}$ for the whole data set is relatively high due to the inclusion of very weak reflections between 7.0 and 5.9 Å resolution. Based on their CC1/2 values (*Karplus and Diederichs 2012*), these weak reflections are contributing significant information and were included in structure refinement. Intensities were converted to structure factors using Ctruncate (*Winn et al., 2011*). Data statistics calculated by Ctruncate, Xtriage, and CNS (*Brunger et al., 1998*) indicated no twinning (*Table 2*).

The structure was determined by molecular replacement with PHENIX (*Adams et al., 2010*) using the Tat:AFF4:P-TEFb complex (PDB ID 4OGR) as the search model. Rigid body refinement in PHENIX resulted in R/R$_{free}$ = 0.36/0.394. The protein complex without TAR was further refined by torsion angle molecular dynamics with deformable elastic network (DEN) restraints γ=0.5, w$_{DEN}$=100 and a slow cooling annealing protocol starting with 3000° K (*Schröder et al., 2010*; *Brunger et al., 2012*)

in CNS (R/ $R_{free}$ = 0.296/0.444). Strong positive density peaks in the Fo-Fc density map (*Figure 4—figure supplement 1*) allowed manual placement of a TAR molecule from the NMR ensemble (PDB ID 1ARJ) (*Aboul-ela et al., 1995*) in Coot (*Emsley and Cowtan 2004*), followed by rigid-body fitting to the Fo-Fc density map using Coot. Alternative orientations of the TAR molecule were also tested but excluded based on poor electron density fits and steric clashes. Rigid body refinement in CNS after TAR placement reduced the $R_{free}$ value by 4% to 0.406. The rigid body refined TAR-complex structure was used as the reference structure in the subsequent DEN refinement. Optimal DEN refinement parameters were determined using the Grid-enabled web service for low-resolution crystal structure refinement (*O'Donovan et al., 2012*) (*Table 2*) Torsion angle molecular dynamics with these optimized DEN refinement parameters, γ=0, w $w_{DEN}$=300, reduced the final R values to R/Rfree=0.223/0.31.5 at 5.9 Å resolution (*Table 2*).

## Hydrogen deuterium exchange

Amide hydrogen-deuterium exchange was initiated by a 20-fold dilution of Tat-AFF4-P-TEFb complex (10 μM stock) into $D_2O$ buffer containing 25 mM Na-HEPES pD 7.4, 0.2 M NaCl and 1 mM DTT at 30°C. The Tat:AFF4:P-TEFb:TAR complex was assembled by incubating Tat:AFF4:P-TEFb (10 μM) with TAR RNA (20 μM) before 20-fold dilution into $D_2O$. After specified time intervals, the exchange was quenched at 0°C with the addition of ice-cold quench buffer (400 mM $KH_2PO_4/H_3PO_4$, pH 2.2). Quenched samples were injected onto an HPLC (Agilent 1100) with in-line peptic digestion and desalting. Desalted peptides were eluted and directly analyzed by an Orbitrap Discovery mass spectrometer (Thermo Scientific). Initial peptide identification was done by running tandem MS/MS experiments. Peptides were identified using a Proteome Discoverer 2.0; Sequest HT (Thermo Scientific) search. Initial mass analysis of the peptide centroids was performed using the software HDExaminer version 1.4.3 (Sierra Analytics) followed by manual verification of every peptide. The deuteron content of the peptic peptides covering Tat, AFF4 and P-TEFb was determined from the centroid of the molecular ion isotope envelope. The deuteron content was adjusted for deuteron gain/loss during digestion and HPLC. Both nondeuterated and fully deuterated Tat:AFF4:P-TEFb complexes were analyzed. Fully deuterated samples were prepared by three cycles of drying and resolubilization in $D_2O$ and 6 M GdnHCl.

## SHAPE analysis of the interactions between the HIV 5'UTR and the Tat:AFF4:P-TEFb complex

The RNA construct contains the 344-nt of the HIV 5'UTR obtained from the Summers lab (*Heng et al., 2012*) with SHAPE handles added at both ends as previously reported (*Bai et al. 2014*). The RNA sample was prepared by in vitro transcription using T7 polymerase with PCR product DNA as templates, and purified using polyacrylamide gel electrophoresis. Purified RNA samples were annealed at 0.1 mg/ml in a buffer containing 50 mM K-HEPES pH 7.5, 200 mM KOAc, and 3 mM $MgCl_2$ by heating at 75°C for 2 min and snap cooling on ice. Before SHAPE reactions, 9 μl of annealed 5'UTR RNA at 0.1 mg/mL was mixed with either 1 μL of the Tat:AFF4:P-TEFb complex to achieve a final stoichiometry of 1:1 (RNA:protein) or matching buffer containing 25 mM HEPES pH 7.3, 0.2 M NaCl, 0.05 M KCl, 3 mM $MgCl_2$, 1 mM TCEP as a control. The resulting mixtures were incubated for 15 min at room temperature. SHAPE probing was performed as previous reported (*Berry et al., 2011*) with 1-methyl-7-mitroisatoic anhydride (1M7) as the 2' hydroxyl-selective electrophile. Raw traces from fragment analysis was analyzed using ShapeFinder (*Vasa et al., 2008*).

## SAXS

SAXS data were collected at the ALS beamline 12.3.1 (Lawrence Berkeley National Laboratory, Berkeley, CA) (*Hura et al., 2009*) at 18°C using a Pilatus 2M detector at 1.5 m sample to detector distance at a wavelength λ=1.127 Å. For each sample, 30 frames were collected with 0.5 s exposure each, and increasing numbers of buffer subtracted data frames were merged until radiation damage effects became noticeable. Sample concentrations for the Tat:AFF4:P-TEFb complex were 3.2 mg/ml, 1.6 mg/ml, and 0.8 mg/ml; those for the Tat:AFF4:P-TEFb complex with TAR were 2 mg/ml, 1 mg/ml, and 0.5 mg/ml. To reduce potential problems with sample aggregation, data were collected within 2 hr from eluting the protein complexes from a S200 size exclusion column, using only the center peak fractions. Buffer-subtracted datasets were merged and analyzed using the program

SCÅTTER (*Förster et al., 2010*). The $D_{max}$ was determined after optimizing the $P(r)$ function using the program SCÅTTER, following recommended procedures. The initial $P(r)$ function was calculated with the default $D_{max}$ of 96 Å. After removing the very low resolution data in the Guinier region, $D_{max}$ was increased until the $P(r)$ function showed no undulations and was positive for all r-values.

Comparison of the data collected at two-fold different concentrations showed some scattering effects due to intermolecular interactions at higher protein concentrations (*Figure 3*). To reduce these effects, we used data collected at the lowest protein concentration for the apo complex. For the TAR complex, we merged the data from the low and high protein concentrations by scaling and merging the middle section of the curves and retaining the data from the lower and higher concentration only for the low and high-resolution data, respectively.

The fit of the computed SAXS profile and χ value of the crystal structure to the experimental SAXS data was calculated using the program FoXS (*Schneidman-Duhovny et al., 2013*, *2010*).

## Modeling of the Tat ARM and CycT1 TRM segments

Tat ARM and CycT1 TRM are disordered protein segments that are key for TAR binding to the Tat: AFF4:P-TEFb complex. These two regions are not well defined in the crystal structure. We used the MODELLER 9.15 loop modeling protocol (*Šali and Blundell, 1993*; *Fiser and Šali, 2003*) to build the missing residues of the Tat ARM segment and to refine the conformation of the CycT1 TRM segment. We added a soft upper bound of 5.2 Å on the distance between Tat residues R52 and R53 and the TAR backbone, based on an NMR spectroscopy study (*Calnan et al., 1991*; *Puglisi et al., 1993*). We constructed 5000 loop models for the Tat ARM and CycT1 TRM segments, followed by hierarchical clustering based on pairwise all-atom RMSD values. We selected the top 5 best scoring clusters based on the DOPE assessment score (*Shen and Sali, 2006*). For each cluster, we used the centroid structure as the representative structure for further analysis. The precision of selected clusters ranges between 2.5 and 6.3 Å.

Prior to obtaining the crystal structure of the TAR-SEC we determined an integrative structure of the complex using HDX, SAXS, and SHAPE data, as well as structural information for the individual components of the TAR complex with Tat:AFF4:P-TEFb. This integrative structure had errors due to incorrect filtering criteria.

## Integrative modeling of TAR binding to the Tat:AFF4:P-TEFb complex without X-ray structure

Before the X-ray structure of Tat:AFF4:P-TEFb:TAR complex was determined, we modeled the complex by integrative structure modeling (*Alber et al., 2007*; *Russel et al., 2012*; *Schneidman-Duhovny et al., 2012*, *2014*). Integrative modeling was based on the atomic structure of the Tat: AFF4:P-TEFb complex (PDB ID 4OGR) (*Schulze-Gahmen et al., 2014*), an atomic structure of TAR (*Schneidman-Duhovny et al., 2010*, *2013*) (PDB ID 1ARJ and 2KX5), a small-angle X-ray scattering (SAXS) profile of the complex, difference HDX data for the apo and TAR complexes, the SHAPE data describing the interactions between the HIV-1 5′UTR and the Tat-AFF4-P-TEFb complex, and previously published biochemical data identifying key residues for the recognition of TAR, including a cross-link between TAR U31 and CycT1 252-261 (*Garber et al., 1998*; *Richter et al., 2002b*). Specifically, we followed an integrative docking protocol (*Schneidman-Duhovny et al., 2012*), using the rigid-body docking program PatchDock (*Schneidman-Duhovny et al., 2005*). We selected only those resulting models whose FoXS computed SAXS profile (*Schneidman-Duhovny et al., 2013*, *2010*) matched the experimental profile with χ < 1.5, followed by hierarchical clustering based on the pairwise all-atom RMSD values. Next, we filtered the remaining representative models further based on the HDX data, requiring that the Solvent Accessible Solvent Area (SASA) of AFF4 residues 57 to 67 was reduced by at least a 10% upon TAR binding. However, in retrospect, this filtering criterion was too strict, resulting in elimination of the correct model of the Tat:AFF4:P-TEFb: TAR complex. The crystal structure revealed that changes in the HDX rates of the AFF4 helical region are in fact caused by a conformational change triggered by binding of TAR rather than direct binding of TAR. The example illustrates the need for a conservative interpretation of experimental data used for structural modeling. It is more prudent to err on the side of caution, and apply spatial restraints that are broader than narrower, resulting in a less precise ensemble of models that still

contains the correct structure as opposed to a more precise ensemble that does not contain the correct structure.

## Luciferase assay

The HeLa-based NH1 cell line containing an integrated HIV-1 LTR-luciferase reporter plasmid (*He et al., 2010*) was transfected with constructs expressing Tat(C22G) only or together with the indicated AFF1 proteins. AFF1 mutants were chosen based on their surface exposed position on the second αhelix in the homologous AFF4 (*Schulze-Gahmen et al., 2014*), with the exception of Val 67, which is only partially exposed. At 48 hr post transfection, total cell lysates were prepared and luciferase activity was measured with kit E1501 from Promega (Madison, WI, USA). An aliquot of cell lysates was analyzed by western blotting with the indicated antibodies to detect the level of the transfected AFF1 protein and the endogenous α-Tubulin.

## Electrophoretic mobility shift assay

Synthetic TAR (nucleotides 18–44) was radioactively labeled as described (*Schulze-Gahmen et al., 2014*). Binding reactions were carried out with 100 pM labeled TAR RNA and Tat:P-TEFb concentrations ranging from 0.009 to 190 nM and five fold molar excess of AFF4 over P-TEFb. Details of the binding conditions and gel chromatography were described previously (*Schulze-Gahmen et al., 2014*). Gel bands were quantified with the program ImageJ (*Schneider et al., 2012*), and dissociation constants were calculated using one-site total binding curves in the program Prizm version 5.0c (GraphPad, La Jolla, CA, USA)

## Acknowledgements

We are grateful to Sarah Keane and Michael Summers for providing 5'UTR RNA and to Chris Jeans (Macrolab at UC Berkeley) for providing equipment for small-scale purification of proteins by size exclusion chromatography. We also thank James Holton for help with X-ray crystal data collection and processing at beamline 8.3.1 at the Advanced Light Source at LBNL. This work was supported by NIH grants P50GM082250 (JHH, JAD, and AS) and NIAID R01AI041757 and R01AI095057 (QZ). SAXS data were collected at the SIBYLS beamline (Lawrence Berkeley National Laboratory) which is funded by DOE/BER contract number DE- AC02-05CH11231 and NIGMS R01GM105404. Beamline 8.3.1 at the Advanced Light Source, LBNL, is supported by the UC Office of the President, Multicampus Research Programs and Initiatives grant MR-15-328599 and the Program for Breakthrough Biomedical Research, which is partially funded by the Sandler Foundation. The Advanced Light Source is supported by the Director, Office of Science, Office of Basic Energy Sciences, of the US Department of Energy under Contract No DE-AC02-05CH11231. This research continues a program begun by the late Thomas Alber, and we dedicate this paper to him.

## Additional information

### Funding

| Funder | Grant reference number | Author |
| --- | --- | --- |
| National Institutes of Health | P50GM082250 | Jennifer A Doudna<br>Andrej Sali<br>James H Hurley |
| National Institute of Allergy and Infectious Diseases | R01AI041757 | Qiang Zhou |
| National Institute of Allergy and Infectious Diseases | R01AI095057 | Qiang Zhou |

The funders had no role in study design, data collection and interpretation, or the decision to submit the work for publication.

### Author contributions

US-G, Conception and design, Acquisition of data, Analysis and interpretation of data, Drafting or revising the article; IE, Analysis and interpretation of data, Drafting or revising the article; GS, YB,

HL, Acquisition of data, Analysis and interpretation of data; DS-D, Carried out integrative modeling, Analysis and interpretation of data; JAD, Conception and design, Analysis and interpretation of data; QZ, AS, JHH, Conception and design, Analysis and interpretation of data, Drafting or revising the article

## Author ORCIDs

Ignacia Echeverria, http://orcid.org/0000-0003-4717-1467

Goran Stjepanovic, http://orcid.org/0000-0002-4841-9949

James H Hurley, http://orcid.org/0000-0001-5054-5445

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
