## [Decision Letter]

Thank you for submitting your article "Insights into HIV-1 proviral transcription from an integrative structure of the P-TEFb:AFF4:Tat:TAR complex" for consideration by *eLife*. Your article has been reviewed by Jonathan Karn (Reviewer #1), James R Williamson (Reviewer #2), and Wes Sundquist, who is a member of our Board of Reviewing Editors, and the evaluation has been overseen Richard Aldrich as the Senior Editor.

The reviewers have discussed the reviews with one another and the Reviewing Editor has drafted this decision to help you prepare a revised submission.

General assessment and substantive concerns:

Building on their earlier X-ray structure of the Tat:AFF4:P-TEFb complex (with the late Dr. Tom Alber), the authors have built experimentally constrained models of Tat:AFF4:P-TEFb in complex with the HIV TAR hairpin. The final ensemble of structures was deduced through a well-validated computational integrative modeling approach that combines SAXS, HDX-MS, and SHAPE data generated in the current study, together with previously published data from NMR studies of TAR in complex with Tat peptides or argininamide, biochemical cross-linking data, and X-ray structures Tat:P-TEFb and the super elongation complex (SEC).

A limitation of the models, which is acknowledged by the authors, is the low resolution of the best fit refined model(s). Nonetheless, the studies dissect for the first time structural mechanisms by which the AFF1/4 SEC scaffold substantially increases the binding affinity of Tat:P-TEFb to TAR. Major findings from the best fit model are:

– Residues on the C-terminal segment of the N-terminal Tat and Cyclin T1 interacting region of AFF4 form (apparent) direct contacts with TAR UCU bulge.

– TAR localizes to a positively charged interface on Tat:P-TEFb:AFF4

– The Tat ARM fits into the TAR major groove with Tat R52 and R53 formingelectrostatic interactions with the TAR UCU bulge. HDX-MS studies suggested that Tat-TAR interactions in the context of SEC are mediated primarily by amino acid side chains.

– Cyclin T1 TRM, previously shown to directly interact with and to be ordered by AFF4, binds TAR in a region spanning the bulge and apical loop.

– Confirmation that the cysteine-rich and hydrophobic core regions of Tat contribute to binding TAR.

Experimental support for the functional interactions of the AFF1/4 scaffold with TAR stems from TAR binding and HIV LTR-luciferase reporter gene assays support the structural findings.

Overall, this is a well-written manuscript provides an insightful addition to long-standing questions about how TAR RNA is recognized and Tat is delivered to the transcriptional machinery. This work will eventually be superseded by X-ray structures of the complex, but it is impossible to anticipate when such structures will become available, and in the meantime, these models provide the best framework available for studying these interactions.

Major issue that should be addressed:

Although SHAPE revealed changes to the A22-U25 bulge as well as the C30-A35 loop region, specific sites of contact on TAR as well as any RNA conformational changes due to SEC binding could not be defined. Moreover, mutations in putative contact residues from AFF1/4 had only modest effects on binding (see below). Previous studies have shown a range of phenotypic effects of point mutations in the UCU bulge and apical loop, as well as transversions base pairs in the TAR stem-loop structure. It would greatly strengthen the argument for direct interactions between AFF1/4 and TAR RNA if mutations in these regions could be shown to selectively limit AFF4 complex binding. If this proves not to be case, then the authors should reconsider their hypothesis that AFF1/4 interacts directly with TAR.

Issues for the authors' consideration:

1) While it is unquestionable from the authors' study that AFF4 robustly increases the affinity of Tat:P-TEFb for TAR, the molecular mechanisms are yet to be properly defined. Point mutations of N-terminal residues on AFF4 that are presumed to be directly involved in TAR binding only elicited a small reduction (>4-fold) in SEC binding to TAR. It is therefore somewhat unclear the extent to which direct AFF4-TAR interactions contribute to the robust strengthening of Tat:P-TEFb binding to TAR. The Results section suggests that the role of direct contacts is secondary to a scaffolding role, but the discussion seems to place a larger importance on the complex formation role of helix α2 of AFF1/4 than the data suggest. Moreover, the authors should at least mention alternative possible explanations, including that they may not have mutated the key contact residues and/or that AFF1/4 may not, in fact, contact TAR directly (i.e., that the modest mutational effects reflect indirect effects).

2) Were replicates performed on the biochemical experiments?

3) While the authors discuss briefly the surprising result that Tat peptides showed little delay in D-exchange, it is unclear how reconcile their reasoning (Results paragraph three) with previously published NMR results showing direct hydrogen bonding (Puglisi et al. 1992).

4) It would be helpful to include a control trace of RT reactions with TAR RNAs in the absence of SHAPE reagents so the reader can assess the presence of natural strong stops which might affect the data analysis (as SI). The data presented are differences, and it is not possible to see positions about which there is no real information.

5) For the SHAPE differences alluded to in subsection “SHAPE mapping of the Tat:AFF4:P-TEFb complex on the HIV 5’UTR”: Were the differences different patterns, or different degree of protection?

6) Please describe how the D_max_ was calculated in SAXS distance distribution plots in the Materials and methods.

7) In the Introduction, it is claimed that more accurate and more precise structures are obtain. This seems unjustified. They are probably more precise, using all information available, but how can better accuracy be claimed, in the absence of independent validation?

8) Introduction: "first experimental structure, albeit at low resolution". Not to quibble, but this language is not precise and does not accurately reflect what was done. The results from this effort are a hybrid model. Some parts of the model are at atomic resolution, and others are relatively unconstrained. "Low resolution structure" doesn't seem quite the right way to describe this. The spirit of the hybrid modeling is very much akin to NMR structure determination, where one never describes the resolution, but rather the RMSD of independent trials. Along these lines, it might be helpful to show superpositions of the resulting models where the alignment is done on only part of the structure (say CycT1:Cdk9), showing the variance of the other regions (as is done for two-domain protein NMR structures for example). This language is also used in the second to last paragraph of the Discussion.

9) Subsection “Integrative modeling of TAR binding to the Tat:AFF4:P-TEFb complex”, second paragraph: 1ARJ is the Abou-ela 1996 structure.

10) If indeed AFF4 causes a 70-fold enhancement in SEC affinity for TAR, why is this not reflected in a HIV transcriptional assay using wildtype Tat? Why did the authors select C22G Tat which would be defective for both P-TEFb and TAR binding based on their structural analysis?

[Editors' note: further revisions were requested prior to acceptance, as described below.]

Thank you for resubmitting your work entitled "Insights into HIV-1 proviral transcription from the structure and dynamics of the Tat:AFF4:P-TEFb:TAR complex" for further consideration at eLife. Your revised article has been evaluated by Richard Aldrich as the Senior Editor, a Reviewing editor, and a reviewer. Since your resubmitted work now includes an X-ray crystal structure, we decided to switch the Reviewing Editor and also consult another reviewer, both with expertise in X-ray crystallography.

The manuscript has been improved but there are some remaining issues that need to be addressed before further consideration, as outlined below:

While it is gratifying that the authors replaced their previous integrative model with a model based on a crystal structure, there are concerns about this new low-resolution crystal structure.

Major comments:

From Table 2 and Figure 4—figure supplement 1, it is difficult to judge if the density shown in Figure 4—figure supplement 1 uniquely determines the proposed position of the RNA. While there is difference electron density at 2σ in this region, how confident are the authors that this is the only possible fit? For example, the model might be positioned backward?

The absence of density of the TAT ARM region is worrisome. The model that the authors propose for the TAT ARM region (Figure 5) suggests multiple contacts between the TAT ARM region and the rest of the complex, so it is surprising that there would be no electron density for it.

As the authors point out, their model of the ARM region differs considerably from that seen in the crystal structure of CycT1-EIAV Tat - EIAV TAR (Anand et al. 2008). Of course, these could be real differences, but the Anand et al. structure is more definitive (at 3.15 Å resolution) compared to the low-resolution model by presented in this work. We do not find the possible explanations given for the differences in structures to be reassuring.

Table 2 indicates a number of possible issues with this crystal structure (low CC1/2 and very high R_merge_ in the highest resolution shell, high R_free_-R_difference_, poor Ramachandran geometry). Although such poor quality indicators may not necessarily indicate incorrectness of the atomic model, they add to the uncertainty of this low-resolution structure.

The authors may have chosen the non-optimal mutations to validate the structure. They mutated individual residues on the exposed surface of AFF4 helix 2 and showed that they had only a small impact on K_d_ be gel shift (one of them had a two-fold effect and another had a 3-fold effect others had no effect). In comparison, complete omission of AFF4 had a 50-fold effect. The concern is that single Ala point mutations are as gentle as one could get, and might quite reasonably have a very modest effect on a direct binding interaction that overlaps this surface compared to complete omission. At the very least, the authors should have made mutations that were trying to falsify the hypothesis - such as radically different substitutions at multiple residues.

In the section "Modeling of Tat ARM and CycT1 TRM", the claim that SAXS data are consistent with the structure is underwhelming. The real question is, what structure would not be consistent with the SAXS data. For example, can they show that the EIAV structure is incompatible with the SAXS data?

In summary, this work could greatly benefit from improved crystals, improved diffraction data collection, and improved atomic model refinement. Improved data and model qualities would lend more confidence into the insights gleaned from this complex. Thus, authors are asked to make a good effort to collect improved diffraction data and to obtain a better atomic model. It is likely that the authors could improve the diffraction quality by optimization of the purification of the complex, crystallization and freezing conditions. Moreover, data collection at a microfocus beamline may improve the limiting resolution of the data.

[Editors' note: further revisions were requested prior to acceptance, as described below.]

Thank you for resubmitting your article "Insights into HIV-1 proviral transcription from integrative structure and dynamics of the Tat:AFF4:P-TEFb:TAR complex" for consideration by eLife. Your article has been reviewed by a Reviewing Editor and Richard Aldrich as the Senior Editor.

We thank the authors for the revision and responses to our concerns. We first would like to clarify that we did not request an "atomic" resolution structure. Rather, we stated that "this work could greatly benefit from improved crystals, improved diffraction data collection, and improved atomic model refinement. Improved data and model qualities would lend more confidence into the insights gleaned from this complex." We of course realize the difficulties in obtaining well-diffracting crystals, but we remain concerned about some of the properties of the diffraction data and associated atomic model. To this end, we examined the PDB validation report that the authors kindly provided. First, according to the report, the geometry is not "good" contrary to what is stated in the Results section of the manuscript. We realize that it is difficult to build a model with good geometry at such low resolution, so this may explain the poor geometry. Moreover, there are some curious abnormalities, for example, the large differences between observed and ideal C=O bond lengths. It is actually extremely important that a model refined against low-resolution data has perfect geometry since the information content of the diffraction data is so low. The authors are incorrect to state that poor geometry at low resolution "is the accepted wisdom in protein crystallography".

The PDB validation report also reveals that four out of five chains that have a rather poor fit to the electron density. Moreover, the report indicates that the diffraction data suffer from a substantial twinning fraction and L-test statistical values that suggest twinning. It is likely that if the crystals were in fact twinned, it might explain the poor fit with electron density as reported by the PDB validation report, the poor fit of TAR into the difference electron density (Figure 4—figure supplement 1), and the absence of density of the Tat ARM. The presence of a substantial twin fraction makes the interpretation of difference maps notoriously difficult due to the inherent introduction of noise by twinning. Moreover, models derived from twinned diffraction data are much more prone to potential model bias than for data without twinning. In relation to this issue, the absence of electron density does not necessarily imply that the particular part of the complex is "flexible", but it may be a consequence of twinning. Thus, the conclusions drawn from this structure for the position and conformation of TAR and the Tat ARM must be viewed with great caution.

The authors argue that their work is not entirely based on the low-resolution crystal structure, and they integrated data from a number of sources to obtain a more complete model. However, we are concerned that this integrated modeling approach did not produce the "correct" structure in the absence of the low-resolution crystal structure, although, according to a previous rebuttal by the authors, it is apparently within a large ensemble structures that was predicted (however, the authors did not provide the entire ensemble). In any case, the crystal structure apparently played a pivotal role to determine then "correct" structure, and therefore must undergo the type of scrutiny that is appropriate for any important crystal structure.

Considering the importance of the crystal structure, the authors are requested to improve the diffraction data and the associated model. Please note that we do not necessarily request higher resolution, but we would ideally like to see a higher quality diffraction data set without twinning, and associated model at the resolution that is claimed here.

---

## [Author Response]

*[…] Overall, this is a well-written manuscript provides an insightful addition to long-standing questions about how TAR RNA is recognized and Tat is delivered to the transcriptional machinery. This work will eventually be superseded by X-ray structures of the complex, but it is impossible to anticipate when such structures will become available, and in the meantime, these models provide the best framework available for studying these interactions.*

We appreciate these comments and note that an x-ray structure (admittedly at just 5.9 Å resolution) has now been determined and added to the manuscript. Despite the limited resolution, the new x-ray structure adds enormously to the confidence level in the RNA positioning, and leads a model that is consistent with the rest of the data.

*Major issue that should be addressed:*

*Although SHAPE revealed changes to the A22-U25 bulge as well as the C30-A35 loop region, specific sites of contact on TAR as well as any RNA conformational changes due to SEC binding could not be defined. Moreover, mutations in putative contact residues from AFF1/4 had only modest effects on binding (see below). Previous studies have shown a range of phenotypic effects of point mutations in the UCU bulge and apical loop, as well as transversions base pairs in the TAR stem-loop structure. It would greatly strengthen the argument for direct interactions between AFF1/4 and TAR RNA if mutations in these regions could be shown to selectively limit AFF4 complex binding. If this proves not to be case, then the authors should reconsider their hypothesis that AFF1/4 interacts directly with TAR.*

The suggested mutations were made. See the new Results subsection “Effects of TAR Mutants on SEC Binding”. They were found to be more permissive for AFF4 binding, the opposite of the prediction from the integrative model. In parallel with these studies, we redoubled efforts to determine the crystal structure of the complex. This was successful, and we now report the crystal structure at 5.9 Å resolution. The structure shows that there are no direct contacts between AFF4 and TAR. AFF4 acts by stabilizing contacts with the TAR central loop, not the bulge, consistent with the effects of the bulge mutants. The integrative model has been removed from the manuscript and replaced with the crystal structure.

We wish to note that the family of models generated in the course of integrative modeling included the correct model. This model was filtered out based on what we now know to be an incorrect criterion that direct contacts be made with AFF4. It is somewhat unusual for such strong protection to be seen for regions that do not make direct contacts. It is tempting to infer that binding sites correspond to regions of high protection since this is often the case. This episode should serve as a caution in the interpretation of HDX protection data in terms of direct vs. indirect binding. A new supplementary text section has been added that describes lessons learned.

*Issues for the authors' consideration:*

*1) While it is unquestionable from the authors' study that AFF4 robustly increases the affinity of Tat:P-TEFb for TAR, the molecular mechanisms are yet to be properly defined. Point mutations of N-terminal residues on AFF4 that are presumed to be directly involved in TAR binding only elicited a small reduction (>4-fold) in SEC binding to TAR. It is therefore somewhat unclear the extent to which direct AFF4-TAR interactions contribute to the robust strengthening of Tat:P-TEFb binding to TAR. The Results section suggests that the role of direct contacts is secondary to a scaffolding role, but the discussion seems to place a larger importance on the complex formation role of helix α2 of AFF1/4 than the data suggest. Moreover, the authors should at least mention alternative possible explanations, including that they may not have mutated the key contact residues and/or that AFF1/4 may not, in fact, contact TAR directly (i.e., that the modest mutational effects reflect indirect effects).*

The new data show that the contribution of AFF4 is large but entirely indirect.

*2) Were replicates performed on the biochemical experiments?*

Yes. Luciferase experiments were repeated 3 times. Gel shift assays were repeated 3 times (2 times in a few cases).

*3) While the authors discuss briefly the surprising result that Tat peptides showed little delay in D-exchange, it is unclear how reconcile their reasoning (Results paragraph three) with previously published NMR results showing direct hydrogen bonding (Puglisi et al. 1992).*

We have gone back to this 1992 manuscript and have found no mention of hydrogen bonds between RNA and main-chain backbone amides. The interactions with arginine side-chain would not be expected to have any effect on the exchange of amide protons.

*4) It would be helpful to include a control trace of RT reactions with TAR RNAs in the absence of SHAPE reagents so the reader can assess the presence of natural strong stops which might affect the data analysis (as SI). The data presented are differences, and it is not possible to see positions about which there is no real information.*

Please see the new Figure 2—figure supplement 1.

*5) For the SHAPE differences alluded to in subsection “SHAPE mapping of the Tat:AFF4:P-TEFb complex on the HIV 5’UTR”: Were the differences different patterns, or different degree of protection?*

Comparison of our Figure 2 and the reference shows differences in pattern and in degree of protection.

*6) Please describe how the D_max_ was calculated in SAXS distance distribution plots in the Materials and methods.*

This is now described in the “SAXS” sub-section in the Material and Methods.

*7) In the Introduction it is claimed that more accurate and more precise structures are obtain. This seems unjustified. They are probably more precise, using all information available, but how can better accuracy be claimed, in the absence of independent validation?*

The word “accurate” was removed, along with the rest of the integrative modeling. In hindsight this word was a particular poor choice, as described in detail in the new supplementary text section.

*8) Introduction: "first experimental structure, albeit at low resolution". Not to quibble, but this language is not precise and does not accurately reflect what was done. The results from this effort are a hybrid model. Some parts of the model are at atomic resolution, and others are relatively unconstrained. "Low resolution structure" doesn't seem quite the right way to describe this. The spirit of the hybrid modeling is very much akin to NMR structure determination, where one never describes the resolution, but rather the RMSD of independent trials. Along these lines, it might be helpful to show superpositions of the resulting models where the alignment is done on only part of the structure (say CycT1:Cdk9), showing the variance of the other regions (as is done for two-domain protein NMR structures for example). This language is also used in the second to last paragraph of the Discussion.*

The integrative model was replaced with a crystal structure, whose resolution at 5.9 Å is, we believe, indisputably “low” by crystallographic standards.

*9) Subsection “Integrative modeling of TAR binding to the Tat:AFF4:P-TEFb complex”, second paragraph: 1ARJ is the Abou-ela 1996 structure.*

Thank you for spotting this mistake. It has been corrected.

*10) If indeed AFF4 causes a 70-fold enhancement in SEC affinity for TAR, why is this not reflected in a HIV transcriptional assay using wildtype Tat? Why did the authors select C22G Tat which would be defective for both P-TEFb and TAR binding based on their structural analysis?*

The reason we used the C22G mutant Tat, which lacks an essential Cys-zinc bridge required for proper folding, is because it is highly sensitive to the AFF-mediated stimulation in the transcription assay. We currently don’t know why WT Tat is far less sensitive in this assay. It is possible that in all the in vivo Tat-transactivation assays we have performed so far, it always involved the introduction of additional AFF1/4 into the cells or knocking down but not completely depleting AFF1/4 from the system. These situations may have made WT Tat, which is incredibly powerful, less sensitive to the stimulatory effect of AFF4 compared to that detected under the highly reductionist gel-shift conditions. We have recently performed a TAR pulldown assay looking for proteins that bind to TAR under Tat (+) and (-) conditions and saw significantly more Tat-mediated loading of the SEC subunits when HeLa nuclear extract (NE) was incubated with TAR RNA. However, when NE prepared from the AFF4-KD cells (~80% KD efficiency) was used in the assay, there was no obvious difference in loading of the rest of the SEC components to TAR. This illustrates that the stimulatory effect of AFF4 is not so easy to detect when analyzed under more complicated situations where there are still some leftover AFF1/4 in the cells.

[Editors' note: further revisions were requested prior to acceptance, as described below.]

*[…] Major comments:*

From Table 2 and Figure 4—figure supplement 1, it is difficult to judge if the density shown in Figure 4—figure supplement 1 uniquely determines the proposed position of the RNA. While there is difference electron density at 2σ in this region, how confident are the authors that this is the only possible fit? For example, the model might be positioned backward?

The TAR orientation is uniquely defined by the difference density. We are including additional figures for 2 alternate TAR orientations.

Alternate orientation #1: Figure 8 shows the density and packing for a TAR rotation, which exchanges the 5’ and 3’ termini of the TAR molecule. The TAR loop in this alternate conformation doesn’t fit the density well, and it clashes with Tat as well as with CycT. In addition, Tat G48, which marks the start of Tat ARM, is positioned next to the minor groove of TAR. This contradicts multiple reports showing Tat ARM interactions with the TAR major groove. Finally, in the orientation described in the manuscript the TAR loop runs parallel to the CycT1 TRM, potentially making contacts over an extended TRM residue range, as would be expected from biochemical data presented in Garber et al 1998. This is not true for the alternate orientation where the TAR loop intersects the CycT TRM in an almost perpendicular orientation.

Alternate orientation #2: Figure 9 shows an alternate orientation with the TAR loop placed where the TAR ends were before, and vice versa. Again, the fit with the electron density is worse, and there are severe steric clashes in the crystal packing between the TAR loops. This orientation also would not agree with any of the published biochemical data nor with the SHAPE data in our paper. Only the orientation described in the manuscript results in good packing and good fit with the density.

In summary, the orientation of TAR is defined by the unbiased crystallographic difference density beyond doubt. We added a statement to this effect to the Discussion. The crystallographic structure is fully consistent with the SHAPE data and the sum of the extensive biological data, and alternative 1 and 2 are not.

Author response image 1.Unbiased difference electron density contoured at 2.4 sigma.**DOI:**
http://dx.doi.org/10.7554/eLife.15910.020

Author response image 2.Unbiased difference electron density contoured at 2.3 sigma.**DOI:**
http://dx.doi.org/10.7554/eLife.15910.021

The absence of density of the TAT ARM region is worrisome. The model that the authors propose for the TAT ARM region (Figure 5) suggests multiple contacts between the TAT ARM region and the rest of the complex, so it is surprising that there would be no electron density for it.

The HDX data address this important question. Figure 1 shows that the Tat peptide 38-57 is 60% H-D exchanged within 10 s, regardless of whether TAR is bound or not. This region of Tat is visualized in the crystal structure (4OGR) through residue 49, and the amides are protected in an helical hydrogen bonds through residue 45. Thus 37% of the peptide is protected from exchange and that matches well with the 40% that is involved in helical amid hydrogen bonding in the crystal structure. This provide strong evidence that the rest of the Tat peptide (46-57, including the ARM) is not involved in hydrogen bonding that would protect that amides from exchange. This essentially rules out the adoption of a helical structure. The absence of protection in this region is most consistent with the ensemble model shown in Figure 5. Wording to this effect has been added to the Discussion.

*As the authors point out, their model of the ARM region differs considerably from that seen in the crystal structure of CycT1-EIAV Tat - EIAV TAR (Anand et al., 2008). Of course, these could be real differences, but the Anand et al. structure is more definitive (at 3.15 Å resolution) compared to the low-resolution model by presented in this work. We do not find the possible explanations given for the differences in structures to be reassuring.*

Anand et al. fused the N-terminus of Tat fused to the C-terminus of the TRM of cycT1. This proved to be an unfortunate choice in retrospect. Subsequent structures of the Tat from the Alber and Tahirov labs (4OGR, 3MI9) showed that the N-terminus is buried in a polar core mostly formed by T1 by also by part of Tat itself. Fusions to the N-terminus will thus disrupt the complex. In the Anand structures of Cyclin T1-Tat (Tat disordered) (2PK2) and Cyclin T1-Tat (Tat core is disordered)-EIAV TAR (2W2H), the T1 TRM occupies the position that was subsequently shown by Tahirov (3MI9) to actually be occupied by the Tat core. Thus it is now clear that the N-terminal fusion of cycT1 to Tat causes the core of Tat to misfold. The region of space that would have been occupied by the Tat core is vacated by Tat and cycT1 TRM moves into this region, causing it to occupy a non-productive conformation.

These structural distortions are consistent with the extremely weak binding of the cycT1- Tat fusion to TAR: at 2 µM TAR, half maximal binding is seen with ~ 3.5 µM fusion construct (Figure 4, Anand, et al. JMB 2007). Compare this to 19 nM for the native PTEFb:Tat complex, a nearly 200-fold loss. Beyond the concern raised by the massive loss of affinity, there were no new mutational studies of the protein contacts were carried out in the Anand study. In other words, the putative protein-RNA contacts in the Anand paper have never been validated.

The difficulties in reconciling the Anand structure with much other data in the field was already discussed by Tahirov et al. & Price, Nature, 2010: “Recently the crystal structure of the fusion of equine cyclin T1 and equine infectious anaemia virus (EIAV) Tat in complex with EIAV TAR RNA was reported. The basic RNA-recognition motif of EIAV Tat adopts a helical structure whose flanking regions interact with an equine cyclin T1 and both proteins coordinate the stem-loop structure of EIAV TAR27. However, the key residues in EIAV TAT and equine cyclin T1 involved in ternary complex formation with TAR RNA, are different in HIV-1 Tat and human cyclin T1. The TAR RNAs from HIV-1 and EIAV have different structures as well. Moreover, the photocross-linking and protein footprinting studies31 show that HIV-1 TAR RNA loop interacts with cyclin T1 residues 252–260, which is far from the EIAV TAR RNA docking site. These differences make it difficult to model HIV-1 TAR binding to the Tat·P-TEFb complex on the basis of the structure of the EIAV complex.”

The working consensus in the field seems to be that the EIAV structure is an artifact of the use of a fusion construct that was never demonstrated to be functional, and indeed, actually has a severe loss of RNA binding function. At best, the structure is irrelevant to the situation with HIV-1. The round of P-TEFb structural papers by Schulze-Gahmen et al eLife 2013 and Gu et al. Cell Cycle 2014, building on Tahirov 2010, do not cite the Anand papers.

Table 2 indicates a number of possible issues with this crystal structure (low CC1/2 and very high R_merge_ in the highest resolution shell, high R_free_-R_difference_, poor Ramachandran geometry). Although such poor quality indicators may not necessarily indicate incorrectness of the atomic model, they add to the uncertainty of this low-resolution structure.

In the overall quality analysis of the structure from the PDB-validation server, our structure is in the 40-50th percentile rank for Rfree, Clashscore, and Ramachandran outliers when compared to other 8.00-3.65 Å structures. Since this is a wide resolution bracket with more structures at the higher resolution end than at the low resolution limit and 5.9 Å is more towards the lower end, these percentile scores seem perfectly acceptable. The CC1/2 value of 0.24 in the high resolution bin corresponds to a true CC (CC*) value of 0.62, which is excellent and is above the cut-off used in the analysis of Karplus & Diederichs, Science 336: 1030-1033, 2012. CC* rather than R_merge_ is currently considered the gold standard in determining data cut-offs. Karplus and Diedrichs stated that “it is justified to include data out to well beyond currently employed cutoff criteria, because the data at these lower signal levels do not degrade the model, but actually improve it.” It is fair to say that this view is the current accepted wisdom in protein crystallography.

*The authors may have chosen the non-optimal mutations to validate the structure. They mutated individual residues on the exposed surface of AFF4 helix 2 and showed that they had only a small impact on K_d_ be gel shift (one of them had a two-fold effect and another had a 3-fold effect others had no effect). In comparison, complete omission of AFF4 had a 50-fold effect. The concern is that single Ala point mutations are as gentle as one could get, and might quite reasonably have a very modest effect on a direct binding interaction that overlaps this surface compared to complete omission. At the very least, the authors should have made mutations that were trying to falsify the hypothesis - such as radically different substitutions at multiple residues.*

It was shown in previous work that AFF4 helix α2 residues M62 and F65 are important for interactions with Tat (Schulze-Gahmen et al., eLIFE, 2014), with the single Ala mutants of their AFF1 cognates reducing transactivation 4-6 fold (and a more substantial ~15 fold for the double mutant). These are very significant effects and they are completely consistent with our current model. The exposed surface of AFF4 mutated in this study was really the last remaining face of the helix that had not already been probed.

*In the section "Modeling of Tat ARM and CycT1 TRM", the claim that SAXS data are consistent with the structure is underwhelming. The real question is, what structure would not be consistent with the SAXS data. For example, can they show that the EIAV structure is incompatible with the SAXS data?*

Our initial integrative model, built without crystallographic data, had a χ =1.4 and eventually proved to be wrong. Similarly, a model composed of human AFF4-P-TEFb with a chimeric HIV/EIAV-Tat (HIV-Tat 1-48, EIAV-Tat 49-57) and HIV-1 TAR positioned as in the Arnon et al structure (pdb 2W2H) resulted in χ =1.3 (Figure 10). Our current model based on crystallographic data, HDX, SHAPE, and SAXS data has a χ value of 0.86, significantly lower than either of the above-mentioned models. The SAXS data taken alone are not proof of the correctness of the model, and we make no claim that they are. They are, as we state, consistent with the model. The SAXS data are substantially more consistent with the current model than with any of the alternatives tested, with a χ < 1, while the alternatives have χ > 1.

Author response image 3.Left panel: TAR-complex from the manuscript (Schulze-Gahmen et al; Figure 5—figure supplement 1).Right panel: Same protein complex with TAR orientated as in EIAV structure and HIV-Tat ARM replaced by EIAV ARM structure.**DOI:**
http://dx.doi.org/10.7554/eLife.15910.022

*In summary, this work could greatly benefit from improved crystals, improved diffraction data collection, and improved atomic model refinement. Improved data and model qualities would lend more confidence into the insights gleaned from this complex. Thus, authors are asked to make a good effort to collect improved diffraction data and to obtain a better atomic model. It is likely that the authors could improve the diffraction quality by optimization of the purification of the complex, crystallization and freezing conditions. Moreover, data collection at a microfocus beamline may improve the limiting resolution of the data.*

We have re-inserted the word “integrative” into the title to highlight that are conclusions are based on the totality of the data. The last sentence of the Abstract has been modified to emphasize that the conclusions are drawn from the totality of the data as processed by integrative modeling, such that the conclusions are not based solely on the crystallography or on any other single technique. The Impact Statement has also been modified to highlight this key point. We would rather not think of this as a low-resolution crystallography paper; our intent was to put together an integrative modeling paper that now goes above and beyond the norm by providing a crystal structure as part of the input data.

The biological problem addressed here is so significant that the original referees and BRE member favorably reviewed the level of insight even at the original low resolution. In order to build confidence in the model, we have now added an intermediate (in a global sense) resolution crystal structure. Since there was already a consensus that even the low-resolution insights were an important advance for the field, it does not seem reasonable raise the bar two notches (from low to intermediate, and then from intermediate to atomic) at the time of the second revision. We make no conclusions that cannot be defended with the available data at the current resolution. In sum, the conclusions are highly significant to the field as judged by a consensus of the original reviewers and editor, and the conclusions are fully supported by the intermediate resolution data provided.

As a practical matter, crystals of the P-TEFb:AFF4:Tat:TAR complex were first obtained two years ago and optimization has been slow and painstaking. Data have been collected on a variety of beamlines. While it is impossible to forecast progress precisely, on the basis of past experience our best estimate is that it will take another two years to obtain an atomic model.

[Editors' note: further revisions were requested prior to acceptance, as described below.]

[…] We thank the authors for the revision and responses to our concerns. We first would like to clarify that we did not request an "atomic" resolution structure. Rather, we stated that "this work could greatly benefit from improved crystals, improved diffraction data collection, and improved atomic model refinement. Improved data and model qualities would lend more confidence into the insights gleaned from this complex."

As a general comment, we realize that there were items in the PDB validation report that could be flagged by a diligent reviewer. Yet we also note that the PDB validation server was not designed with the special situation of low resolution structures in mind. The method used by this server to detect twinning is not robust at low resolution, and the electron density fits used the automated map calculation does not used the low resolution-optimized scaling procedure from the CNS DEN method that we followed. As we document below, our structure is in all respects up to par for a 5.9 Å structure composed of substructures that had been individually well-refined against high resolution diffraction or NMR data.

We of course realize the difficulties in obtaining well-diffracting crystals, but we remain concerned about some of the properties of the diffraction data and associated atomic model. To this end, we examined the PDB validation report that the authors kindly provided. First, according to the report, the geometry is not "good" contrary to what is stated in the Results section of the manuscript. We realize that it is difficult to build a model with good geometry at such low resolution, so this may explain the poor geometry.

There was no manual model building involved in our study other than rigid-body docking of an RNA conformer from 1ARJ into density. Refinement was done using the DEN algorithm in CNS with tight coupling of our structure to the reference structure, which avoids over-fitting, as documented by the reasonable free R-factor. Apart from protein-RNA contacts, and the CO bond length addressed below, the model quality is essentially that of the parent structures, which were determined at high resolution (protein) and by solution NMR (RNA).

Moreover, there are some curious abnormalities, for example, the large differences between observed and ideal C=O bond lengths.

The CNS parameter file that we used for phosphothreonine had a main-chain CO bond length parameter that was 0.1 Å too long. The file was read in after the others and overwrote the standard parameters such that all main chain CO bonds were set 0.1 Å too long. This has nothing to do with over-fitting, and at 5.9 Å resolution, the 0.1 Å coordinate error is meaningless. At any rate, this mistake

has been corrected and the PDB deposition replaced. There is no impact on any other aspect of the structure, maps, R-factor, or interpretation. A new validation report is attached.

*It is actually extremely important that a model refined against low-resolution data has perfect geometry since the information content of the diffraction data is so low. The authors are incorrect to state that poor geometry at low resolution "is the accepted wisdom in protein crystallography".*

The geometry for every part of every molecule was tightly restrained to the starting model using the DEN methodology, with a Cα r.m.s.d. of 0.4 Å.

The PDB validation report also reveals that four out of five chains that have a rather poor fit to the electron density.

The maps used in our analysis were scaled in CNS using the DEN script. The fit quality of the appropriately scaled map generated by CNS is excellent and exactly what would be expected at 5.9 Å. See the new Figure 4—figure supplement 2. The automated map calculation used by the PDB validation server does not use the same scaling procedure and so produces a much poorer map, and corresponding RSCC and RSR score values that do not reflect the actual fit to the CNS DEN map. A file of real space correlation coefficient (RSCC) and R-factor (RSR) values generated by CNS is attached. Most RSCC correlation values are well above 0.9.

Moreover, the report indicates that the diffraction data suffer from a substantial twinning fraction and L-test statistical values that suggest twinning. It is likely that if the crystals were in fact twinned, it might explain the poor fit with electron density as reported by the PDB validation report, the poor fit of TAR into the difference electron density (Figure 4—figure supplement 1), and the absence of density of the Tat ARM. The presence of a substantial twin fraction makes the interpretation of difference maps notoriously difficult due to the inherent introduction of noise by twinning.

The structure is not twinned. We inquired with the PDB and they stated that their automated analysis of twinning carried out by the validation server is not reliable at 5.9 Å resolution. The PDB ran additional checks that show there is no twinning. See also the output from CTRUNCATE and CNS (Figure 11), all demonstrating zero twinning.

It is not unusual for the initial difference density for a large missing substructure to be imperfect just after molecular replacement and the first round of refinement. Many authors don’t show these maps for this reason, but we consider it the proper thing to do. The 2Fo-Fc map shown in the new Figure 4—figure supplement 1 shows an excellent fit for the TAR RNA. Of course, the 2Fo-Fc map is, like any such map, potentially subject to model bias. We showed the difference map as a main figure despite its aesthetic imperfection because it is completely unbiased and the TAR RNA is by far the largest feature in the difference map.

There is in fact a density feature at the expected ARM position, however it is not possible to model it with confidence and so did not seem appropriate to speculate about it in the manuscript. The wording about the ARM density at the bottom of pg. 7 is slightly more hedged in the revised manuscript.

Author response image 4.Output of CTRUNCATE**DOI:**
http://dx.doi.org/10.7554/eLife.15910.023

Moreover, models derived from twinned diffraction data are much more prone to potential model bias than for data without twinning.

The only phases used for modeling RNA were calculated with the RNA absent throughout its previous refinement history. No rebuilding of protein was carried out at all, and no rebuilding of the RNA was carried out subsequent to its initial docking into the density. Model bias is not relevant in the context of this structure.

*In relation to this issue, the absence of electron density does not necessarily imply that the particular part of the complex is "flexible", but it may be a consequence of twinning. Thus, the conclusions drawn from this structure for the position and conformation of TAR and the Tat ARM must be viewed with great caution.*

*The authors argue that their work is not entirely based on the low-resolution crystal structure, and they integrated data from a number of sources to obtain a more complete model. However, we are concerned that this integrated modeling approach did not produce the "correct" structure in the absence of the low-resolution crystal structure, although, according to a previous rebuttal by the authors, it is apparently within a large ensemble structures that was predicted (however, the authors did not provide the entire ensemble). In any case, the crystal structure apparently played a pivotal role to determine then "correct" structure, and therefore must undergo the type of scrutiny that is appropriate for any important crystal structure.*

*Considering the importance of the crystal structure, the authors are requested to improve the diffraction data and the associated model. Please note that we do not necessarily request higher resolution, but we would ideally like to see a higher quality diffraction data set without twinning, and associated model at the resolution that is claimed here.*

The conclusion regarding the extended nature of the ARM and the exposure of its main-chain is drawn from the HDX data, not from the absence of density. There actually is some limited density, as mentioned above. We use the term “extended” rather than “flexible”. It may ultimately prove at higher resolution that the ARM is at least partly ordered via side-chain interaction. What is clear from the HDX is that at least the main-chain amide groups of the ARM must be exposed to solvent, which rules out a helical conformation for the ARM.